# Combining transgenesis with paratransgenesis to fight malaria

Wei Huang[1], Joel Vega-Rodriguez[2], Chritopher Kizito[1], Sung-Jae Cha[1], Marcelo Jacobs-Lorena[1]*

[1]Department of Molecular Microbiology and Immunology, Malaria Research Institute, Johns Hopkins Bloomberg School of Public Health, Baltimore, United States; [2]Laboratory of Malaria and Vector Research, National Institute of Allergy and Infectious Diseases, National Institutes of Health, Rockville, United States

**Abstract** Malaria is among the deadliest infectious diseases, and *Plasmodium*, the causative agent, needs to complete a complex development cycle in its vector mosquito for transmission to occur. Two promising strategies to curb transmission are transgenesis, consisting of genetically engineering mosquitoes to express antimalarial effector molecules, and paratransgenesis, consisting of introducing into the mosquito commensal bacteria engineered to express antimalarial effector molecules. Although both approaches restrict parasite development in the mosquito, it is not known how their effectiveness compares. Here we provide an in-depth assessment of transgenesis and paratransgenesis and evaluate the combination of the two approaches. Using the Q-system to drive gene expression, we engineered mosquitoes to produce and secrete two effectors – scorpine and the MP2 peptide – into the mosquito gut and salivary glands. We also engineered *Serratia*, a commensal bacterium capable of spreading through mosquito populations to secrete effectors into the mosquito gut. Whereas both mosquito-based and bacteria-based approaches strongly reduced the oocyst and sporozoite intensity, a substantially stronger reduction of *Plasmodium falciparum* development was achieved when transgenesis and paratransgenesis were combined. Most importantly, transmission of *Plasmodium berghei* from infected to naïve mice was maximally inhibited by the combination of the two approaches. Combining these two strategies promises to become a powerful approach to combat malaria.

*For correspondence:
ljacob13@jhu.edu

Competing interest: The authors declare that no competing interests exist.

## Editor's evaluation

This article provides convincing evidence that combining transgenic with paratransgenic approaches can provide a useful tool to reduce malaria transmission by *Anopheles* mosquitoes. The study provides a series of well-designed experiments that will be of interest to malaria and vector control specialists as well as to a broader audience interested in genetic manipulation of insects and paratransgenesis.

## Introduction

An estimated 241 million malaria cases and 627,000 malaria deaths worldwide were reported in 2020 (**WHO, 2021**). Whereas world malaria incidence has declined by 27% during the first 15 years of this century, in the last 4 years it declined by less than 2%, indicating that current interventions to control this deadly disease are waning (**WHO, 2021**). The development of innovative approaches to reduce this intolerable burden is sorely needed.

The strategy of targeting the mosquito to fight malaria is based on two premises: (1) the mosquito is an obligatory vector for parasite transmission and (2) strong bottlenecks limit parasite development

in the mosquito and during transmission to the mammalian host (*Smith et al., 2014*). The mosquito acquires the parasite when it bites an infected individual. Of the large number of gametocytes (~$10^3$) ingested by the mosquito, only a few (single digits) ookinetes succeed in traversing the mosquito gut and differentiate into oocysts, defining the first strong bottleneck (*Wang and Jacobs-Lorena, 2013*). Each oocyst produces thousands of sporozoites, a good proportion of which invade the salivary glands, where they are stored. Only a small number of these sporozoites (on the order of 1% of total salivary gland content) are delivered when an infected mosquito bites a new individual, defining a second strong bottleneck (*Vanderberg, 1977*).

Since the early demonstration that mosquitoes can be engineered to be refractory to the parasite (*Ito et al., 2002*), the effectiveness of this approach has been robustly demonstrated in the laboratory by simultaneous expression of multiple effector genes (genes that stop parasite development without affecting the mosquito vector) (*Wang et al., 2012*; *Dong et al., 2020*). The major current challenge is to devise means to introduce the genes that confer refractoriness into mosquito populations. This will most likely be achieved by use of CRISPR/Cas9 gene drives (*Carballar-Lejarazú et al., 2020*; *Quinn and Nolan, 2020*). In addition to technical aspects, topics to be resolved include regulatory and ethical issues related to the release of genetically modified organisms in nature.

An independent approach to suppress the mosquito vectorial capacity is to express effector genes from symbiotic bacteria rather than from the mosquito itself, an approach referred to as paratransgenesis. Paratransgenesis has the advantage that the bacteria occur in the mosquito gut in large numbers, in close proximity to the most vulnerable parasite forms. Since the early demonstration of the effectiveness of paratransgenesis to contain the spread of *Trypanosoma cruzi*, the causative agent of Chagas disease, by the *Rhodnius prolixus* vector (*Durvasula et al., 1997*), this approach has been developed for suppressing the mosquito's ability to vector the malaria parasite (*Wang et al., 2012*; *Yoshida et al., 1999*; *Shane et al., 2018*; *Riehle et al., 2007*). As is the case for gene drive, the mosquito symbiont *Serratia AS1* can spread through mosquito populations and be engineered to secrete effector proteins (*Wang et al., 2017*).

This work addresses two unanswered questions: (1) which of the two genetic approaches – transgenesis and paratransgenesis – is the most effective, and (2) can the two approaches be combined to enhance the effectiveness of the intervention? We use transgenic mosquitoes engineered to express effector genes in the midgut and/or salivary glands and *Serratia* bacteria engineered to express the effector genes. We measured the ability of these two strategies, individually and in combination, to inhibit malaria parasite transmission.

## Results

### Generation of *Anopheles stephensi* mosquitoes expressing antimalaria effectors

To constitutively and robustly express antimalaria effector proteins in the midgut and salivary glands of *An. stephensi* mosquitoes, we used the QF-QUAS binary expression system previously adapted for expression in *Anopheles gambiae* (*Potter et al., 2010b*; *Riabinina et al., 2016*). We constructed two 'driver' mosquito lines that express the QF transcription factor, one driven by the constitutive salivary gland-specific anopheline antiplatelet protein (AAPP) promoter (*Shen and Jacobs-Lorena, 1998*) and the other driven by the constitutive midgut-specific peritrophin 1 (Aper1) promoter (*Abraham et al., 2005*; *Yoshida and Watanabe, 2006*; *Figure 1A*). We also constructed a third 'effector' mosquito line that encodes two parasite inhibiting factors (MP2 and scorpine) downstream of the QUAS promoter and driven by the QF transcription factor (*Figure 1A*). Crossing this effector line with either or both driver lines leads to the salivary gland and/or midgut expression of parasite-inhibiting factors. The midgut peptide 2 (MP2) dodecapeptide, identified from a phage display screen, binds tightly to the mosquito midgut epithelium, and inhibits *Plasmodium falciparum* invasion with high efficiency (*Vega-Rodríguez et al., 2014*), whereas the scorpion (*Pandinus imperator*) peptide scorpine lyses malaria parasites without affecting mosquito fitness (*Conde et al., 2000*; *Gao et al., 2010*). Each of the three constructs also expresses YFP (yellow eyes, salivary gland QF driver), dsRed (red eyes, midgut QF driver), or CFP (blue eyes, QUAS effector) fluorescent selection markers (*Figure 1A*).

Two midgut driver lines (Mg1 and Mg2), two salivary gland driver lines (Sg1 and Sg2), and two effector lines (E1 and E2) were obtained. Transgenic mosquitoes were screened by fluorescence

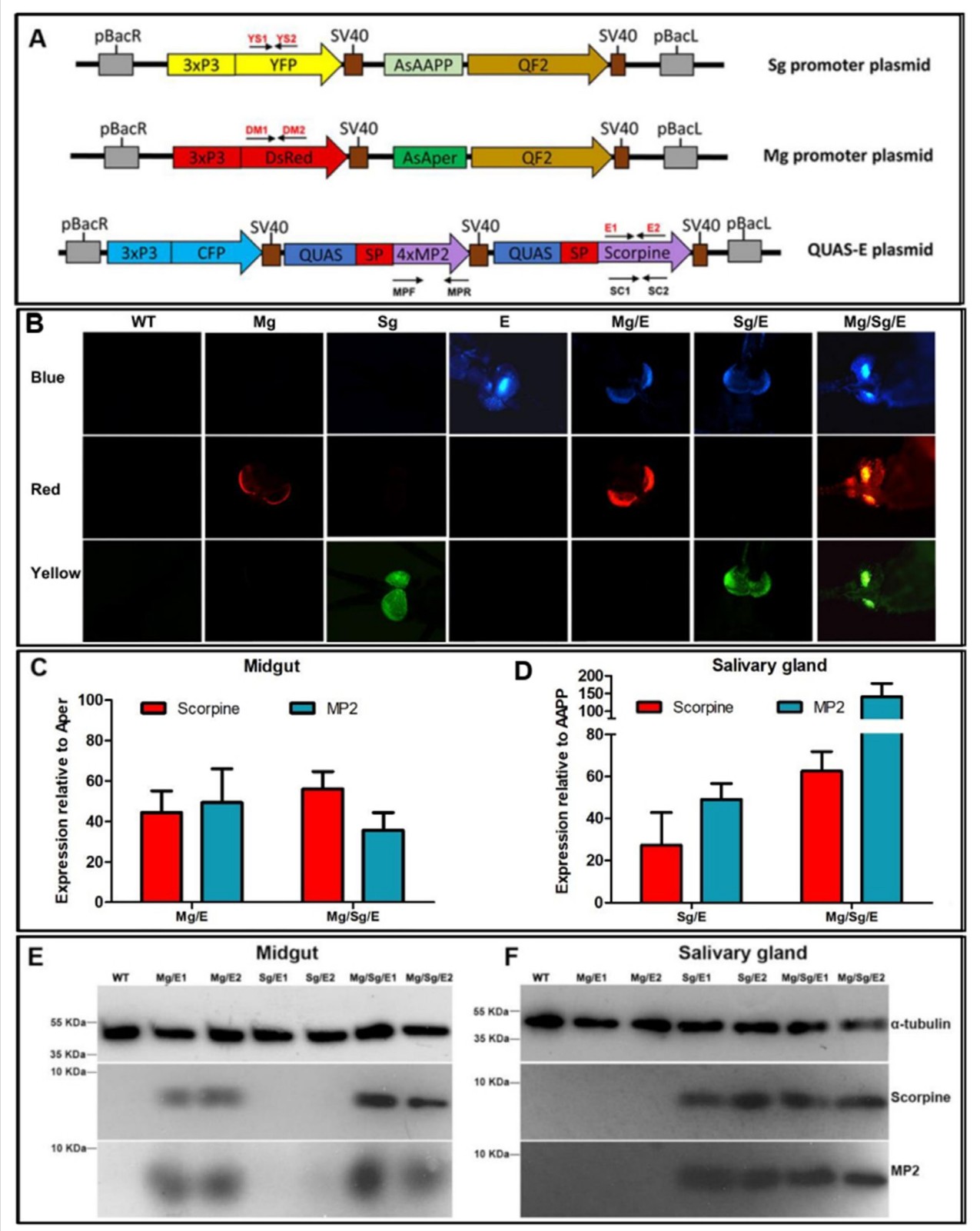

**Figure 1.** Tissue-specific expression of effector genes in *An. stephensi* transgenic mosquitoes. (**A**) Diagram of the salivary gland (Sg) and midgut (Mg) driver constructs expressing the QF2 transcription factor and the effector (E) constructs expressing the MP2 and scorpine effector proteins under control of the QUAS promoter. Each construct also includes sequences encoding a yellow (YFP), red (DsRed), or blue (CFP) fluorescent protein under the control of the 3xP3 eye promoter. pBac: piggyBac inverted terminal repeats; SV40: transcription terminator sequence; SP: *An. stephensi* carboxypeptidase

*Figure 1 continued on next page*

*Figure 1 continued*

signal peptide. Primers used for validation of insertion into mosquito lines (*Appendix 1—figure 1* and *Appendix 1—table 7*) are indicated in red font. Primers used for qRT-PCR are indicated in black font (*Appendix 1—table 7*). (**B**) Detection of fluorescent eye markers in wild-type (WT) and transgenic mosquitoes carrying different combinations of midgut driver (Mg), salivary gland driver (Sg), and effector (**E**) sequences. (**C, D**) Tissue-specific expression of MP2 and scorpine mRNA in transgenic mosquitoes quantified by qRT-PCR in the midgut relative to the endogenous Aper mRNA (**C**) and the salivary glands relative to the endogenous AAPP mRNA (**D**). Mosquito rpS7 was used as a reference. Data pooled from three independent biological replicates. Statistical analysis was determined by Student's *t*-test. (**E , F**) Immunoblotting showing MP2 (6.17 kDa) and scorpine peptide (8.46 kDa) protein expression in midgut and salivary gland lysates from WT and transgenic lines. α-Tubulin was used as a loading control. E1 and E2 refer to independent mosquito transgenic lines. Antibodies used are shown to the right of (**F**).

The online version of this article includes the following source data for figure 1:

**Source data 1.** Source data of *Figure 1B, C, D, E and F*.

microscopy (*Figure 1B*), and plasmid insertion was verified by PCR (*Appendix 1—figure 1*). The position of genome integration was determined for each parental line by splinkerette PCR (*Potter and Luo, 2010a*) and sequencing (*Appendix 1—table 1*). All the parental lines, except for Sg2, have insertions in intergenic regions. Two of the three Sg2 insertions are in intergenic regions, and one in the open-reading frame of the gamma-glutamyltranspeptidase gene (ASTE010947). Each transgenic line was propagated for over 10 generations, discarding at each generation mosquitoes not displaying the correct combination of fluorescent eyes. After the 10th generation, 20 transgenic female mosquitoes from each line were mated with wild-type (WT) males, and 100% the offspring showed the same fluorescence as the parent females, consistent with all transgenic lines being homozygous (*Appendix 1—table 2*).

## Quantification of effector mRNA and protein expression

Using reverse transcription quantitative polymerase chain reaction (RT-qPCR), we compared abundance of the endogenous mosquito Aper and AAPP transcripts with the abundance of effector transcripts originating from the same promoters but driven by the Q-system. Transcripts derived from the Q-system were substantially higher. In the midgut, the scorpine transcript was between 44- (p<0.01) and 56-fold (p<0.01) higher than that of the endogenous Aper mRNA and the MP2 transcript abundance was between 49- (p<0.001) and 36-fold (p<0.01) higher, depending on the transgenic line (*Figure 1C*, *Appendix 1—table 3*). In the salivary glands, scorpine transcript abundance varied between 27- (p<0.05) and 63-fold (p<0.01) higher and MP2 transcript between 49- (p<0.001) and 140-fold (p<0.01) higher than that of the endogenous AAPP mRNA, depending on the transgenic line (*Figure 1D*, *Appendix 1—table 4*). Moreover, in the absence of a driver, transgene expression in 'E' effector mosquitoes (see *Figure 1A*) was undetectable (*Appendix 1—table 3* and *Appendix 1—table 4*). These results attest to the high effectiveness of the Q-system in enhancing tissue-specific transgene expression. Western blot analysis using anti-MP2 and anti-scorpine antibodies confirmed the tissue-specific expression of the MP2 (6.17 kDa) and scorpine (8.46 kDa) proteins (*Figure 1E and F*).

## Mosquito fitness is not affected by effector gene expression and paratransgenesis

To determine whether DNA integration or antimalaria effector expression affects mosquito fitness, we analyzed the survival of WT, parental transgenic (Mg, Sg, and E), and transgene-expressing mosquitoes. No significant longevity differences were detected for any female (*Appendix 1—figure 2A*) or male (*Appendix 1—figure 2B*) transgenic mosquitoes compared to WT. Next, we determined the fecundity (number of laid eggs) and fertility (percentage of hatched eggs) of WT, parental, and antimalaria transgenic lines. Mosquitoes from all parental and antimalaria-expressing transgenic lines showed no difference in fecundity when compared to WT mosquitoes (*Appendix 1—figure 2C*). As for fertility, no significant differences were detected for the Mg and Sg/E lines when compared to WT, while only marginal differences were detected for the Sg, E, Mg/E, and Mg/Sg/E lines (2.0, 3.1, 2.0, and 2.0% reduction, respectively) (*Appendix 1—figure 2D*).

To determine whether transgenesis or antimalaria gene expression in the midgut and/or in the salivary glands affects blood feeding, we quantified the proportion of mosquitoes that take a blood meal (feeding rate) and the amount of blood ingested per mosquito. We found no significant differences

(*Appendix 1—figure 2E and F*), suggesting that neither transgenesis nor antimalaria gene expression affects blood ingestion. In further experiments, we found that there are no differences in lifespan, fertility, or fecundity among WT, transgenic, and (transgenic plus paratransgenic) mosquitoes (*Appendix 1—figure 3A–D*).

In summary, our data show that transgenesis, antimalaria gene expression in the midgut and/or salivary glands, and paratransgenesis do not impair mosquito survival, fecundity, fertility (only minor differences), and blood feeding under laboratory conditions.

## Effector-expressing *Serratia* populate the mosquito reproductive organs and are transmitted vertically and horizontally

The *Serratia AS1*-multi strain that produces and secretes multiple effector proteins, including scorpine and MP2 (*Wang et al., 2017*), was tested here. This strain can populate mosquitoes and be

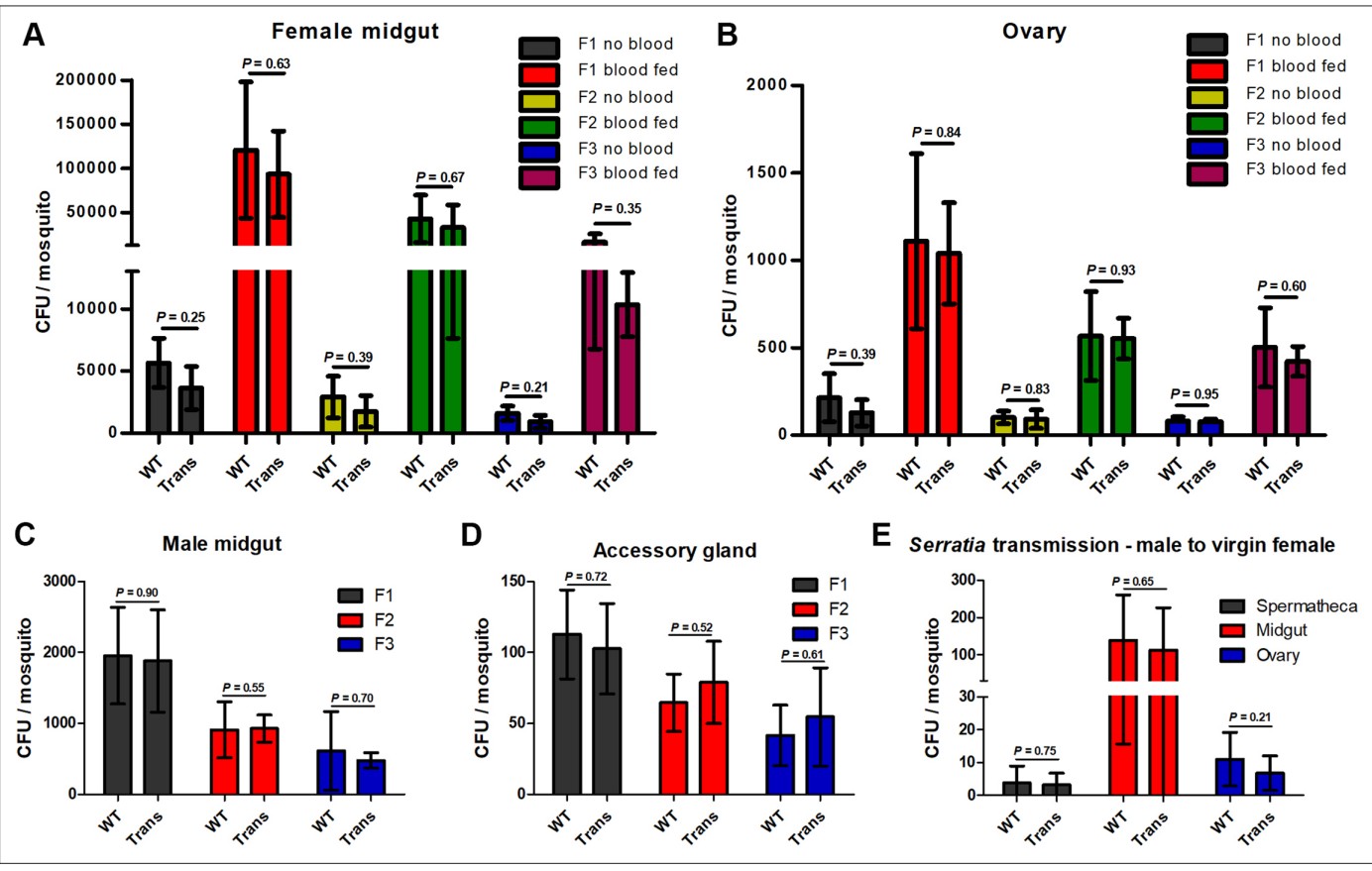

**Figure 2.** *Serratia AS1*-multi-effector bacteria persist through multiple mosquito generations. A total of 100 wild-type (WT) or transgenic (Trans) virgin females that had been fed with *AS1*-multi-effector bacteria were placed in a cage with 100 WT or transgenic virgin males (not fed with bacteria) and allowed to mate. Mosquitoes were then fed blood and allowed to lay eggs. These eggs were allowed to hatch and reared to adults following standard protocol (F1). The F1 mosquitoes were propagated through two additional generations (F2 and F3) without providing additional genetically modified bacteria. At each generation, 10 mosquitoes were dissected, and bacterial load was determined by plating serial dilutions of tissue homogenates on apramycin and ampicillin agar plates and counting colonies. (**A**) Colony-forming units (CFUs) per female midgut fed or not on blood. (**B**) CFUs per female ovary fed or not on blood. (**C**) CFUs per male midgut. (**D**) CFUs per male accessory gland. Data pooled from three independent experiments. (**E**) *Serratia* horizontal (sexual) transmission. Newly emerged virgin male adult mosquitoes were fed on 5% sugar solution containing $10^7$ *AS1*-multi-effector bacteria/ml and then allowed to mate with virgin females. Three days later, 10 females were assayed for the presence of *Serratia AS1* by plating spermatheca, midgut, and ovary homogenates on apramycin and ampicillin agar plates and counting colonies. Trans: Mg/Sg/E transgenic mosquitoes. Error bars indicate standard deviation of the mean. Data pooled from three independent biological experiments. Statistical analysis was determined by Student's *t*-test.

The online version of this article includes the following source data for figure 2:

**Source data 1.** 'Figure 2ABCD-source data.xlsx' is the original colony-forming unit (CFU) data for *Figure 2A–D*; 'Figure 2ABCD-source data.pzf' shows that *Figure 2A–D* were generated with GraphPad Prism.

transmitted through multiple generations (*Wang et al., 2017*). We fed WT and Mg/Sg/E transgenic female mosquitoes with *Serratia AS1*-multi bacteria and quantified their ability to colonize different mosquito organs, and to be transmitted along consecutive mosquito generations. We found that *Serratia AS1*-multi equally populate WT and transgenic mosquito midguts, ovaries, and accessory glands and are transmitted for at least three generations (*Figure 2A–D*). Moreover, we found that WT and transgenic male mosquitoes colonized with *Serratia AS1*-multi transferred the bacteria horizontally (sexually) to virgin WT and transgenic female mosquitoes (*Figure 2E*). Horizontal transfer did not take place when male mosquitoes were placed with *mated* females, showing that transfer occurs during copulation (*Appendix 1—table 5*; female mosquitoes mate only once in their lifetime). These results suggest that recombinant *Serratia* AS1 can effectively populate transgenic mosquitoes and be transmitted through multiple generations.

The possibility that *Serratia* bacteria carried by the mosquito are incorporated into its salivary glands and then delivered to the mammalian host when it bites raises concern. To address this possibility, mosquitoes previously fed with fluorescently labeled *Serratia* were allowed to feed on blood using a membrane feeder. The remaining blood in the feeder was collected, grown overnight in LB medium, and plated. No bacteria were detected (*Appendix 1—figure 4*). We note that this assay is very sensitive as one life bacterium in the blood is expected to result in abundant growth during the overnight incubation.

## Transgenic and paratransgenic expression of effector genes inhibits *Plasmodium* development in the mosquito

Mosquitoes carrying or not experimental bacteria were fed with the same *P. falciparum* infectious blood. Infections were followed by verifying the presence of the effector proteins in the blood meal

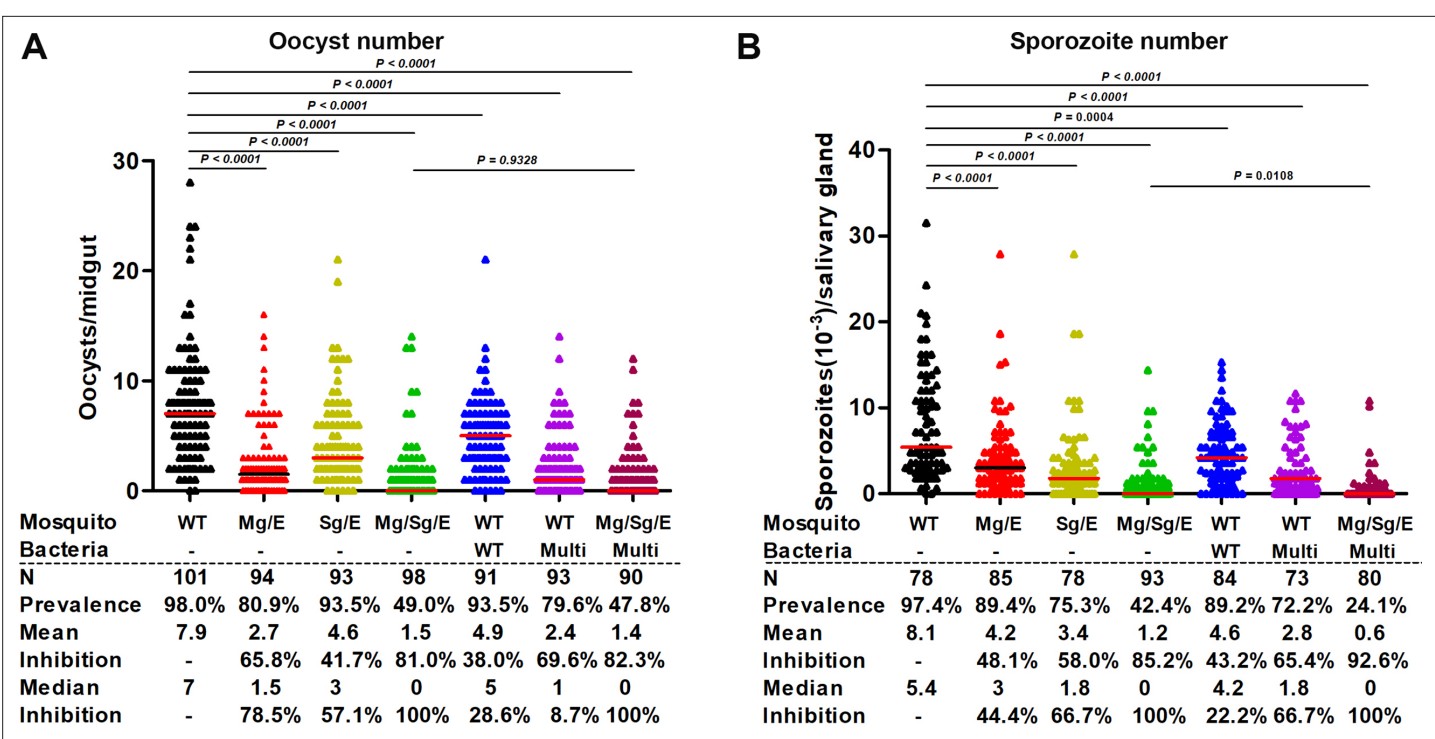

**Figure 3.** Transgenesis and paratransgenesis strongly impair *Plasmodium* development. Two-day-old *An. stephensi* mosquitoes were fed (or not) overnight with wild-type or recombinant *Serratia* AS1-multi bacteria, as indicated. After 48 hr, all mosquito groups were fed on the same *P. falciparum* gametocyte culture and midgut oocyst number was determined on day 7 (**A**) and salivary gland sporozoite number was determined on day 14 (**B**) post-feeding. Horizontal lines represent median oocyst or sporozoite number. Data pooled from three independent biological experiments. Statistical analysis was done by Mann–Whitney *U*-test. 'multi': *Serratia* AS1-multi bacteria expressing multiple effectors; N: number of mosquitoes assayed; Prevalence: proportion of mosquitoes carrying one or more parasite.

The online version of this article includes the following source data for figure 3:

**Source data 1.** Source data of *Figure 3A and B*.

of the experimental mosquitoes. In paratransgenic mosquitoes, only the multi-effector protein was detected, while in the (paratransgenic + transgenic) mosquitoes scorpine and MP2 peptides originating from the bacteria were detected, in addition to the multi-effector protein expressed by the mosquitoes (*Appendix 1—figure 6*). As shown in *Figure 3*, expression of effector molecules in the midgut or in the salivary glands of transgenic mosquitoes significantly reduced parasite burden, whereas concomitant effector expression in both organs reduced burden to the greatest extent (81.0 and 85.2% inhibition of mean oocyst and sporozoite numbers, respectively). Effector-expressing recombinant bacteria also significantly reduced parasite burden in WT mosquitoes (69.6 and 65.4% inhibition of oocyst and sporozoite numbers, respectively). As found previously (*Wang et al., 2017*), WT bacteria also inhibited to some extent oocyst formation in WT mosquitoes. Importantly, combining mosquito transgenesis with paratransgenesis led to the strongest inhibition of parasite development. Oocyst prevalence was reduced from 98.0% to 49.0% for transgenic-only mosquitoes and to 47.8% when transgenesis and paratransgenesis were combined. Sporozoite prevalence was reduced from 97.4% to 42.4% for transgenic-only mosquitoes and to 24.1% when transgenesis and paratransgenesis were combined.

In an attempt to determine which parasite stage parasite was affected by effector expression, we measured ookinete formation in the midgut of mosquitoes fed with an infectious blood meal (*Appendix 1—figure 5*). For all transgenic and paratransgenic combinations, ookinete formation was strongly inhibited, suggesting that the effector molecules affect the early parasite stages in the mosquito midgut.

The results so far suggest that the ability of (transgenic + paratransgenic) mosquitoes to transmit the parasite may be strongly impaired, a hypothesis that was tested next.

## Malaria transmission is maximally impaired by combining transgenesis and paratransgenesis

To investigate the ability of mosquitoes to transmit the parasite from an infected to a naïve animal, we challenged naïve mice with the bite of mosquitoes that had ingested the same infectious blood meal. Four mosquito groups were investigated: (1) WT mosquitoes, (2) WT mosquitoes carrying *Serratia* AS1-multi (paratransgenic), (3) transgenic mosquitoes that express effectors in the midgut and salivary glands (transgenic), and (4) transgenic mosquitoes carrying *Serratia* AS1-multi (paratransgenic + transgenic) (*Figure 4A*). All four mosquito groups fed on the same *Plasmodium berghei*-infected mouse, ensuring that all mosquitoes ingested blood with the same parasitemia. At 21–23 days post-feeding, after the mosquito salivary glands were populated by sporozoites, either three (*Figure 4B*) or five (*Figure 4D*) mosquitoes were randomly selected and allowed to bite naïve mice. For each experiment, salivary gland sporozoite numbers were determined (*Figure 4C and E*).

When mice were challenged with the bite of three WT mosquitoes (three independent experiments with five mice each), 100% became infected (half-infection time = 5.5 ± 0.5 days) (*Figure 4B*) and their salivary glands had a median 8400 sporozoites (*Figure 4C*). With mosquitoes carrying *Serratia* AS1-multi (paratransgenic), 26.7% of the mice were not infected (half-infection time 7.1 ± 0.7 days), and their salivary glands had a median of 2100 sporozoites (74% lower than WT mosquitoes). With transgenic mosquitoes, 67% of the mice were not infected, and their salivary glands had a median of 900 sporozoites (92% lower than WT mosquitoes). With (paratransgenic + transgenic) mosquitoes, 93% mice were not infected, and their salivary glands had a median of zero sporozoites (100% lower than WT mosquitoes).

When mice were challenged with the bite of five WT mosquitoes (*Figure 4D and E*), only 1 mouse out of 15 (6.7%) did not get infected (half-infection time = 5.6 ± 0.7 days). With paratransgenesis, 26.7% of the mice were not infected (half-infection time = 8.3 ± 1.0), with transgenesis 47% of the mice were not infected (half-infection time = 10.1 ± 1.0 days) and with (paratransgenesis + transgenesis) 80% of the mice were not infected. The salivary gland sporozoite number (*Figure 4E*) was similar to that observed for experiments with three mosquito bites (*Figure 4C*).

In summary, our data shows that transgenic and paratransgenic expression of effector molecules are both effective in impairing transmission, but that the combination of the two strategies is considerably more effective. An even higher effectiveness is expected from the bite of one infected mosquito, which is the most likely scenario in the field.

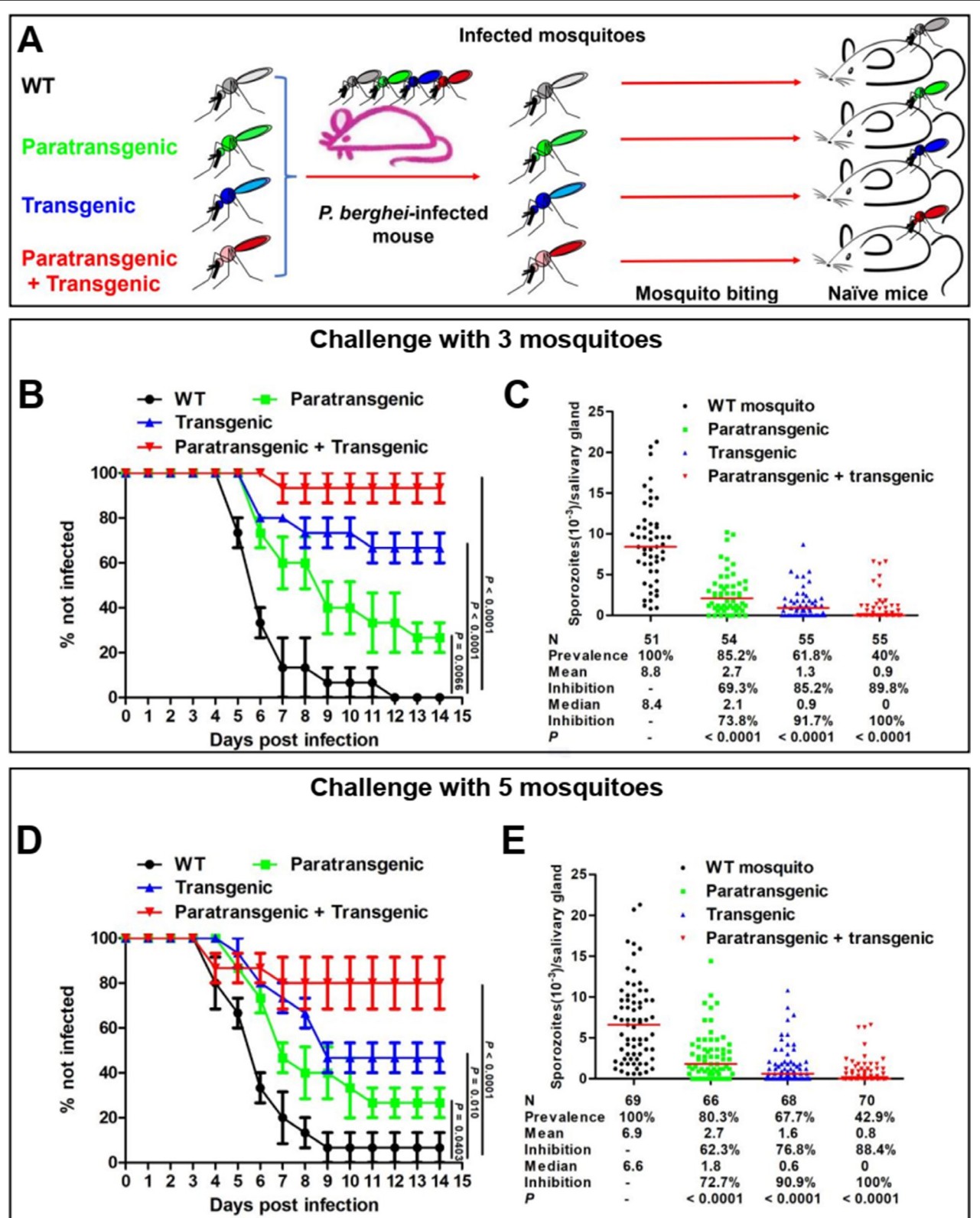

**Figure 4.** Transgenesis and paratransgenesis inhibit *P. berghei* transmission by mosquitoes from infected to naïve mice. (**A**) Experimental design. Wild-type (WT), paratransgenic, transgenic, and (paratransgenic + transgenic) mosquitoes were fed on the same *P. berghei*-infected mouse, assuring that all mosquitoes ingested infected blood with the same parasitemia. After 21~23 days, when sporozoites had reached the salivary glands (**C, E**), three (**B**) or five (**D**) mosquitoes were randomly selected and allowed to bite naïve mice. The parasitemia of these mice was followed for 14 days. Data pooled

*Figure 4 continued on next page*

*Figure 4 continued*

from three independent experiments, each using five mice per challenged group for a total of 15 mice. Transgenic mosquitoes express effectors in both midgut and salivary glands. Statistical analysis was determined by log-rank (Mantel–Cox) test (**B, D**) or Mann–Whitney *U*-test (**C, E**).

The online version of this article includes the following source data for figure 4:

**Source data 1.** Source data of *Figure 4B, C, D and E*.

## Discussion

In this study, we report the development of a new class of transgenic mosquitoes driven by the Q-system. We also assess the effectiveness of transgenesis and paratransgenesis, individually or in combination, in thwarting *Plasmodium* parasite transmission. Notably, effector mRNA abundance was about 50 times higher than that of the endogenous genes, consistent with the high effectiveness of the Q-system in *Drosophila* (*Potter et al., 2010b*). Some QF toxicity was reported when the Q-system was first used in *Drosophila* (*Potter et al., 2010b*; *Riabinina et al., 2016*). Of note, expression of neither the QF transcription factor nor the antimalaria effectors affected mosquito longevity, blood meal uptake, or offspring production under laboratory conditions. Additional experiments are required to test the fitness of these transgenic mosquitoes under field conditions.

We selected two potent effector molecules, MP2 and scorpine, to block the development of *Plasmodium* in the mosquito. MP2 is a 12-amino-acid peptide that likely targets a midgut receptor for ookinete traversal (*Vega-Rodríguez et al., 2014*), and scorpine is an antimicrobial toxin hybrid between a cecropin and a defensin that lyses *Plasmodium* ookinetes (*Conde et al., 2000*). Scorpine expressed by the entomopathogenic fungus *Metarhizium* in the mosquito hemocoel strongly inhibits (~90%) salivary gland sporozoite numbers (*Fang et al., 2011*). Transgenic mosquitoes expressing this effector in the salivary glands were also highly effective in reducing sporozoite numbers (this work). Furthermore, expression of both effector genes in the midgut and the salivary glands led to a much stronger decrease of salivary gland sporozoite numbers than the expression of the effectors in either of these organs alone. A number of effectors have already been individually tested in paratransgenesis experiments (*Wang et al., 2017*). Going forward, the combination of different effectors and the use of mosquitoes and bacteria expressing different effector sets should be explored to achieve maximum blocking activity.

That expression of antimalaria effectors in the salivary glands inhibited oocyst development in the midgut (Sg/E; *Figure 3A*) is most likely explained by the fact that mosquitoes ingest saliva with the blood meal, in this way incorporating effector proteins into the blood bolus (*Luo et al., 2000*). A similar phenomenon was also observed in a recent report showing that human PAI-1 expressed in salivary glands was ingested together with the saliva and inhibited oocyst formation (*Pascini et al., 2022*). In our study, we confirmed the presence of the multi-effector protein, in addition to the AAPP salivary gland protein (control), in the midguts of Sg/E mosquitoes that express the multi-effector protein only in the salivary glands (*Appendix 1—figure 7*). Scorpine has been shown to be nontoxic to insect cells (*Carballar-Lejarazú et al., 2008*), whereas MP2 toxicity was not determined previously. Further studies are needed to determine whether the delivery of these molecules by the mosquito bite could induce physiological responses. As effector molecules in the mosquito saliva are injected into the host dermis during blood feeding, the introduction in the field of transgenic mosquitoes that produce nonhuman proteins in their saliva needs to be considered with much caution.

Experiments seeking evidence for possible bacteria transmission with a mosquito bite yielded negative results (*Appendix 1—figure 4*), suggesting that mosquitoes cannot inoculate the bacteria while feeding on a host. It was previously shown that secretion of effector proteins by recombinant *Pantoea* (*Wang et al., 2012*), *Serratia* (*Wang et al., 2017*), or *Asaia* (*Shane et al., 2018*) bacteria into the midgut inhibits *Plasmodium* development and that WT *Serratia* AS1 is transmitted from one mosquito generation to the next (*Wang et al., 2017*). What was not known is whether engineering *Serratia* to produce and secrete large amounts of proteins would affect their fitness and ability to be transmitted. Our experiments showed that the engineered *Serratia* were efficiently transmitted from one mosquito generation to the next, a result that bodes well for the implementation of the paratransgenesis strategy in the field. For introduction of bacteria in the field, we envision placing around villages, cotton baits soaked with a mosquito attractant dissolved in sugar and containing suspended bacteria. This is similar to how we introduce bacteria into mosquitoes in the laboratory. The continuous

introduction of bacteria into the local mosquito population, combined with the seeding the breeding sites when female mosquitoes lay eggs covered with bacteria (*Wang et al., 2017*), is expected to compensate for the decrease of transmission through mosquito generations.

This project was based on two basic premises: (1) transgenesis and paratransgenesis are not mutually exclusive and (2) both strategies result in impairment of parasite development in the mosquito. As such, our experiments addressed the question of whether a combination of the two strategies would result in enhanced transmission-blocking effectiveness. The combination of transgenesis and paratransgenesis greatly reduced parasite development in the mosquito, and most importantly, it resulted in a high-level reduction of transmission from an infected to a naïve mouse compared to the individual interventions. When naïve mice were bitten by three (transgenic + paratransgenic) mosquitoes, 93% of the mice (14 out of 15) did not develop an infection compared with 100% infection when mice were bitten by WT mosquitoes. In the field, where the density of infected mosquitoes is low even in high-transmission areas, it is unlikely that people will be consecutively bitten by more than one infected mosquito, and protection from transmission is expected to be very high. For translating these findings to the field, the testing of different combinations of effectors, both for transgenesis and paratransgenesis, may further improve the effectiveness of the approach.

Whereas both transgenesis and paratransgenesis have been shown to be highly effective in a lab setting, the challenge will be to implement these new strategies in the field. In addition to address regulatory and ethical issues connected with the release of recombinant organisms in nature, a major technical issue to be solved is how to introduce the blocking transgenes into mosquito populations in the field. In this respect, CRISPR/Cas9 technology has afforded the development of promising gene drive systems focused on population suppression or population modification strategies (*Scudellari, 2019*; *Nolan, 2021*; *Simoni et al., 2020*; *Adolfi et al., 2020*). Population reduction leaves an empty biological niche that upon cessation of reduction pressure will result in recolonization by the same or other mosquito species. In contrast, population modification results in a more stable state, with a biological niche occupied by mosquitoes that are poor transmitters. Similarly, efficient spread of recombinant bacteria into mosquito populations has been demonstrated in a laboratory setting (reference *Wang et al., 2017* and this work), indicating a promising path toward the field implementation of the most efficient (transgenesis + paratransgenesis) strategy. The recent finding that a naturally occurring and nonmodified *Serratia* can spread through mosquito populations while strongly suppressing *Plasmodium* development (*Gao et al., 2021*) significantly increases the feasibility of moving a paratransgenesis-like approach into the field as it bypasses concerns relating to the release of genetically modified organisms in nature. Notably, transgenesis and paratransgenesis are not envisioned to be implemented by themselves. Both are compatible with current vector and malaria control measures such as insecticide-based mosquito control, mass drug administration, and vaccines, and their added implementation promises to substantially enhance the effectiveness of intervention of disease transmission.

In summary, we show that the Q-binary system to express anti-*Plasmodium* effectors in the mosquito is highly efficient. We also show that in addition to inhibiting parasite development, recombinant *Serratia* AS1 is horizontally and vertically transmitted across multiple mosquito generations, which is a bacteria counterpart of gene drive. A major conclusion of this work is that the combination of transgenesis with paratransgenesis provides maximum parasite blocking activity and has high potential for fighting malaria.

## Materials and methods
### Mosquitoes rearing and parasite culture

*An. stephensi* Nijmegen strain (*Feldmann and Ponnudurai, 1989*) and *An. stephensi* transgenic lines were reared as previously described *Huang et al., 2020*. For fitness evaluation, the mosquitoes were fed on Swiss Webster mice.

Female *An. stephensi* were infected with *P. falciparum* gametocyte cultures via membrane feeding. *P. falciparum* NF54 gametocytes were produced according to *Tripathi et al., 2020*. Briefly, the parasites were maintained in O+ human erythrocytes using RPMI 1640 medium supplemented with 25 mM HEPES, 50 mg/l hypoxanthine, 25 mM $NaHCO_3$, and 10% (v/v) heat-inactivated type O+ human serum (Interstate Blood Bank, Inc) at 37°C and with a gas mixture of 5% $O_2$, 5% $CO_2$, and balanced $N_2$.

For feeding, 14–17-day-old mature gametocytes were pelleted by centrifugation (5 min, 2500 × $g$), resuspended with O+ human RBC to 0.15–0.2% gametocytemia, and diluted to 40% hematocrit with human serum. All manipulations were done maintaining the cultures, tubes, and feeders at 37°C.

## Plasmid constructs

The pXL-BACIIECFP-15XQUAS-TATA-MP2-SV40-15XQUAS-TATA-scorpine-SV40 containing the MP2 and Scorpine expression cassette and the ECFP gene under the eye-specific promoter 3xP3 was used to generate the parental QUAS-[MP2 + scorpine] effector lines (*Appendix 1—table 6*). The coding DNA for MP2-SV40-15XQUAS-TATA-Scorpine was synthetized by GeneScript (*Appendix 1—figure 8*). The sequence was amplified using primers MP2-ScopineF and MP2-ScopineR (*Appendix 1—table 7*), and In-Fusion-cloned into plasmid pXL-BACIIECFP-15XQUAS-TATA-SV40 (*Wang et al., 2017*) previously linearized with XhoI.

The pXL-BACII-DsRed-AsAper-QF2-hsp70 containing the QF2 transcription factor under the control of the midgut specific AsAper promoter and the DsRed marker driven by the eye-specific promoter 3xP3 was used to generate the parental Mg-QF driver line. The AsAper promoter (1.5 kb) (*Appendix 1—figure 8*) was PCR-amplified from *An. stephensi* gDNA with primers MgPF and MgPR (*Appendix 1—table 7*). The PCR product was In-Fusion-cloned into plasmid pXL-BACII- DsRed-QF2-hsp70 (*Potter et al., 2010b*) previously linearized with XhoI.

The pXL-BACII-YFP-AsAAPP-QF2-hsp70 containing the QF2 transcription factor under the control of the midgut specific AsAAP promoter and the YFP marker driven by the eye-specific promoter 3xP3 was used to generate the parental Sg-QF driver lines. The YFP coding sequence was amplified using primers YFPF and YFPR (*Appendix 1—table 7*, *Appendix 1—figure 8*). The PCR product was In-Fusion-cloned into plasmid pXL-BACII-DsRed-QF2-hsp70 previously digested with ApaI and NotI to produce plasmid pXL-BACII-YFP-QF2-hsp70. The AsAAPP promoter consisting of a 1.7 kb upstream of the start codon (*Yoshida and Watanabe, 2006*) was PCR-amplified from *An. stephensi* gDNA using primers SgPF and SgPR (*Appendix 1—table 7*, *Appendix 1—figure 8*). The PCR product was In-Fusion-cloned into plasmid pXL-BACII-YFP-QF2-hsp70 previously linearized with XhoI.

## Generation of transgenic mosquitoes

The plasmid constructs were microinjected into *An. stephensi* embryos as described (*Volohonsky et al., 2015*). Briefly, transformation plasmids were purified using the EndoFree Maxi Prep Kit (QIAGEN) and resuspended in injection buffer (0.1 mM NaHPO$_4$ pH 6.8 and 5 mM KCl) at a concentration of 250 ng/μl for the transformation plasmid and 200 ng/μl for the helper plasmid encoding the transposase. The plasmid mix was injected into *An. stephensi* embryos using a FemtoJet Microinjector (Eppendorf). Third-instar larvae of G$_0$ survivors were screened for transient expression of the 3xP3-dsRed marker (red eyes), 3xP3-YFP marker (yellow eyes), and 3xP3-CFP marker (blue eyes). Adults obtained from the fluorescent marker screening were crossed to WT mosquitoes to generate independent transgenic lines. The data for these injections are summarized in *Appendix 1—table 8*.

For each of the parental transgenic lines, splinkerette PCR (*Potter and Luo, 2010a*) and PCR sequencing were used to determine the transgene insertion site into the *An. stephensi* genome. Two rounds of amplifications were conducted with 1X Phusion High-Fidelity PCR Master Mix with HF Buffer (Thermo Fisher Scientific). The primers used are shown in *Appendix 1—table 7*. The amplified PCR products were resolved in a 1.5% agarose gel stained with ethidium bromide, and the amplified DNA bands from the 5′ and 3′ ends were individually excised and purified with QIAquick Gel Extraction Kit (QIAGEN). Purified PCR products were cloned into pJET1.2/blunt plasmid (Thermo Fisher Scientific) and transformed into NEB 5-alpha Competent *Escherichia coli* (High Efficiency, Thermo Fisher Scientific). Plasmids were isolated from individual colonies and sequenced with the universal primers pJET12F and pJET12R (Eurofins). The sequences were aligned to the *An. stephensi* genome using VectorBase and NCBI BLAST to identify the location of transgene insertion sites (*Appendix 1—figure 1*).

To obtain homozygous lines, each transgenic line was propagated for more than 10 generations, discarding at each generation mosquito larvae not displaying the expected fluorescent eyes. To verify homozygosity of the transgenic lines, 10 females of each line were mated with 10 WT male mosquitoes, fed blood, and eggs were collected and reared to larvae. The larvae were individually inspected

for expression of the fluorescent protein marker(s). Absence of the expected fluorescence would indicate that the parent female was heterozygous for this dominant marker.

To induce midgut- or salivary gland-specific expression of MP2 and scorpine, QF driver lines were crossed to QUAS-[MP2 + scorpine] effector lines. The offspring of each cross was selected by the specific combination of eye fluorescence reporters (*Figure 1B*).

## Quantitative reverse transcription polymerase chain reaction (qRT-PCR)

Tissue-specific expression of MP2 and scorpine mRNAs in *An. stephensi* transgenic lines was evaluated by RT-PCR. Salivary glands and midguts were dissected from female mosquitoes in ice-cold 200 µl TRIzol (Thermo Fisher Scientific). Total RNA was extracted according to TRIzol manufacturer's protocol, resuspended in RNAse-free water, and treated with RQ1 RNase-Free DNase (Promega, Madison, WI). After RNA quantification using a DeNovix DS-11 spectrophotometer, first-strand cDNA was synthesized for each sample using Superscript III (Invitrogen) with random hexamers (Invitrogen) and 500 ng of total RNA per sample. cDNA was treated with RNase H (New England Biolabs) for 10 min at 37°C and stored at –70°C until use. The cDNA was used as template in PCR reactions containing the Taq 2X Master Mix (New England Biolabs) and 5 µM of MP2- and scorpine-specific primers (*Appendix 1—table 7*). Amplification of S7 ribosomal mRNA was used as reference (*Zhang et al., 2011*). PCR conditions were 1 hot start at 95°C for 30 s; 35 cycles of denaturation at 95°C for 30 s, annealing at 56°C for 30 s, and elongation at 68°C for 30 s; followed by a final extension at 68°C for 5 min; and 4°C indefinitely.

## Mice immunization

Scorpine epitope (CEKHCQTSGEKGYCHGT, the N-terminus was conjugated to KLH) and MP2 epitope (ACYIKTLHPPCS, the N-terminus was conjugated to KLH) were synthesized by Peptide 2.0 Inc About 6–8-week-old C57BL/6 mice were immunized with 20 µg (50 µl) purified antigen in PBS using Addavax (Invivogen, San Diego, CA) as the adjuvant. A total of 50 µl adjuvant was mixed with 50 µl antigen, and the mixture was administered intramuscularly in both anterior tibialis muscles (50 µl per leg). Mice were immunized twice at 2-week intervals. Serum was collected 14–21 days after administration of the last booster (*Cha et al., 2018*).

## Commercial antibodies

Rabbit anti-α-tubulin was purchased from Sigma (Cat# SAB3501072) and goat anti-rabbit IgG HRP-conjugated and goat anti-mouse IgG HRP-conjugated were purchased from Cell Signaling (Cat# 7076S).

## Western blotting

MP2 and scorpine protein synthesis in midgut and salivary glands of the transgenic lines was evaluated by Western blot. Also, 5 midguts and 10 salivary glands were dissected in PBS and placed in microtubes containing RIPA Buffer (Thermo Fisher Scientific), 1% Halt Protease Inhibitor Cocktail (Thermo Fisher Scientific), and 0.1 mM PMSF (Sigma-Aldrich). Samples were homogenized and stored at –70°C. An equivalent of 0.25 midgut and 5 salivary glands were resolved in a NuPAGE 10% Bis-Tris Protein Gel (Invitrogen) under reducing conditions and transferred to a PVDF membrane Invitrogen Power Blotter Select Transfer Stacks. After the transfer, the membrane was washed with TBST 1% (Sigma-Aldrich), incubated with blocking buffer (5% milk powder in TBST 1%) overnight at 4°C, and probed with mouse anti-MP2 or anti-scorpine at a 1:1000 dilution in TBST 1% overnight at 4°C. The membrane was washed and incubated with an anti-mouse HRP-linked antibody (Cell Signaling) at a 1:10,000 dilution in TBST 1% for 2 hr at room temperature. Detection was done with the SuperSignal West Dura Extended Duration Substrate Chemiluminescent Substrate (Thermo Fisher Scientific) and imaged using an Azure Imager c600 (Azure Biosystems).

## Mosquito survival, fecundity, and fertility

To measure mosquito survival, 2-day-old adult male and female mosquitoes (n = 100) were separately placed in a cage with cotton pads soaked in 10% sucrose solution and kept in the insectary. Female mosquitoes were allowed to blood feed on an anesthetized mouse for 30 min and allowed to lay eggs. Mortality of female and male mosquitoes was monitored three times per week. The differences

among the survival curves (three independent replicates) were analyzed with log-rank (Mantel–Cox) test using the WT as controls.

To assess fecundity (number of laid eggs) and fertility (percentage of hatched eggs), 2-day-old adult females were blood-fed on anesthetized mice for 30 min. Only fully engorged females were used for these experiments. Two days after blood feeding, 20 females were individually placed in 50 ml tubes containing a small cup with filter paper soaked in 2 ml of distilled water as a oviposition substrate. After 3 days, the filter papers with eggs were removed, and the number of eggs per mosquito was counted using a dissecting microscope. After counting, the eggs were placed in paper cups with 50 ml of distilled water to allow hatching. Fertility was determined as the number of larvae divided by the total number of eggs. Fecundity and fertility of the transgenic lines were compared to WT mosquitoes, and all the experiments were repeated for a total of three biological replicates.

To measure mosquito survival with paratransgenesis, *Serratia* bacteria were administered overnight to female *An. stephensi* with a cotton pad soaked with a 5% sucrose solution containing $10^7$ bacteria/ml or no bacteria, and 2 days later, female mosquitoes were allowed to blood feed on an anesthetized mouse for 30 min and allowed to lay eggs. Mortality of female and male mosquitoes was monitored three times per week. The differences among the survival curves (three independent replicates) were analyzed with log-rank (Mantel–Cox) test using the WT as controls.

To assess fecundity (number of laid eggs) and fertility (percentage of hatched eggs) with paratransgenesis, *Serratia* bacteria were administered overnight to female *An. stephensi* with a cotton pad soaked with a 5% sucrose solution containing $10^7$ bacteria/ml or no bacteria, and 2 days later, female mosquitoes were allowed to blood feed on an anesthetized mouse for 30 min and allowed to lay eggs. Only fully engorged females were used for these experiments. Two days after blood feeding, 20 females were individually placed in 50 ml tubes containing a small cup with filter paper soaked in 2 ml of distilled water as an oviposition substrate. After 3 days, the filter papers with eggs were removed, and the number of eggs per mosquito was counted using a dissecting microscope. After counting, the eggs were placed in cups with 50 ml of distilled water to allow hatching. Fertility was determined as the number of larvae divided by the total number of eggs. Fecundity and fertility of the transgenic lines were compared to WT mosquitoes, and all the experiments were repeated for a total of three biological replicates.

## Quantification of blood uptake

The amount of blood ingested by *An. stephensi* transgenic mosquitoes was determined by measuring the amount of protein-bound heme detected in the mosquito midgut after a blood meal (*Alves E Silva et al., 2021*). Transgenic and WT mosquitoes were fed with a 1:1 mixture of plasma and RBCs (Interstate Blood Bank Inc) using membrane feeders. After feeding, the midguts of 10 fully engorged females were dissected and homogenized individually in 1 ml of distilled water. Unfed mosquitoes were used as the negative control. Protein-bound heme (410 nm) was measured for each individual midgut with a Versa max microscope Reader and recorded with Softmax pro 5.3. Readings were compared among the groups using Student's *t*-test.

## Bacteria administration to *An. stephensi* mosquitoes

After culturing at 28°C overnight, bacteria were washed with sterile PBS and resuspended to a final concentration of $10^9$/ml. After a 3 hr starvation, mosquitoes were fed overnight on $10^7$ CFU bacteria (*AS1*-multi, apramycin resistance) per ml of 5% sugar. Mosquitoes were surface sterilized with cold 75% ethanol for 3 min and washed three times with sterile PBS. Midguts were dissected under sterile conditions at different time points before and after a blood meal and homogenized in sterile PBS. Bacterial number was determined by plating tenfold serial dilutions of the homogenates on LB agar plates containing 50 µg/ml apramycin and ampicillin (bacteria from noninfected mosquitoes cannot grow on LB agar plates containing 50 µg/ml apramycin and ampicillin) and incubating at 28°C for 24 hr.

## Effect of bacteria on mosquito infection by *P. falciparum*

*Serratia* bacteria were administered overnight to female *An. stephensi* with a cotton pad soaked with a 5% sucrose solution containing $10^7$ bacteria/ml or no bacteria, and 2 days later, allowed to feed on *P. falciparum* NF54 gametocyte-containing blood as described (*Fang et al., 2011*). Engorged

mosquitoes were kept at 27°C and 80% relative humidity. Midguts were dissected in 1× PBS at 7 days post-infection, stained with 0.1% mercurochrome, and oocysts were counted. Salivary glands from mosquitoes were dissected at 14 days post-infection and individually homogenized on ice in 30 µl of PBS using a disposable pestle. The homogenate was centrifuged at 2000 rpm for 10 min to pellet tissue debris. Then, 10 µl of the suspension was placed in a Neubauer counting chamber, waiting for at least 5 min to allow sporozoites to sediment to the bottom of the chamber. Sporozoites were counted using a Leica phase-contrast microscope. Parasite numbers among control and experimental groups were compared using the nonparametric Mann–Whitney test (GraphPad Prism).

*Serratia* bacteria were administered overnight to female *An. stephensi* with a cotton pad soaked with a 5% sucrose solution containing $10^7$ bacteria/ml or no bacteria, and 2 days later, allowed to feed on *P. falciparum* NF54 gametocyte-containing blood as described. The number of ookinetes in the midgut were determined 22 hr post feeding. Each midgut was disrupted in 20 µl PBS by pipetting and transferred to 8-well slides, 1 midgut/well. After drying at room temperature, the samples were fixed in 80% methanol for 15 s and allowed to dry at room temperature. Samples were blocked with 5% BSA for 1 hr at room temperature and then incubated for 1 hr at room temperature with Pfs25 antibody (Mab4b7) (*Barr et al., 1991*) in blocking buffer (1:250–1:500 dilution). After three washes with PBS for 5 min, samples were incubated for 1 hr at room temperature with Alexa Fluor 488 goat anti-mouse IgG (Life Technologies, Cat# A11001) in blocking buffer (1:250–1:500 dilution). After washing three times with PBS for 5 min, slides were allowed to dry and covered with cover slips. Ookinetes were counted using fluorescent microscopy. Parasite numbers among control and experimental groups were compared using the nonparametric Mann–Whitney test (GraphPad Prism).

### Effect of bacteria on mosquito infection by *P. berghei*

Bacteria were cultured overnight in LB medium and washed three times with sterile PBS. Two-day-old mosquitoes were fed overnight on a cotton pad soaked with a 5% sucrose solution containing or not $10^7$ bacteria/ml. Two days later, mosquitoes were fed on a *P. berghei*-infected mouse (1–2% of parasitemia and 1 exflagellation per 10 fields). Unfed mosquitoes were removed, and fully engorged mosquitoes were provided with 5% (wt/vol) sterile sucrose solution and maintained at 19°C and 80% relative humidity. Midguts were dissected on day 12 after the blood meal, stained with 0.1% (wt/vol) mercurochrome for determining oocyst load. Salivary glands were dissected at 21 days post-infection for sporozoite determination. Transgenic and WT mosquitoes were simultaneously fed on the same *P. berghei*-infected mouse to assure that control and experimental mosquitoes ingested the same number of parasites.

### *Serratia* vertical, venereal, and transstadial transmission

To test vertical transmission, *AS1*-multi were introduced into 2-day-old adult female mosquitoes by feeding them overnight on a cotton pad moistened with 5% sterile sucrose containing $10^7$ bacteria/ml. Two days later, mosquitoes were fed on a healthy mouse and were then allowed to lay eggs on a damp filter paper in individual oviposition tubes. Eggs were collected into a tube containing 300 µl sterile 1× PBS and homogenized. The bacterial load was determined by plating tenfold serial dilutions of the egg homogenates on LB agar plates containing 50 µg/ml of apramycin and ampicillin and incubating the plates at 28°C for 24 hr for colony counting. Rearing of larvae to adults followed standard protocol. A total of 10 male and 10 female adults were sampled and examined by plating adult midgut homogenates on LB agar plates containing apramycin and ampicillin. To test the efficiency of *Serratia* transmission through multiple generations, the mosquitoes were reared without providing additional *Serratia* AS1 and maintained for three consecutive generations. At each generation, 10 female and male adults were sampled for examining the presence of *AS1*-multi effectors.

For male-to-female venereal transmission tests, *Serratia* were introduced into newly emerged virgin male mosquitoes by feeding them overnight on a cotton pad moistened with 5% sugar solution containing $10^7$ bacteria/ml. Twenty *Serratia*-carrying males were then allowed to mate with 20 three-day-old virgin females. Three days after mating, 10 females were sampled and examined for bacteria in the female midgut, ovary, and spermatheca.

## Transmission from infected to naïve mice

Transgenic and WT mosquitoes were simultaneously fed on the same *P. berghei*-infected mouse and unfed or partially fed mosquitoes were removed. Midguts from a small number of mosquitoes were dissected at 12 days post-feeding to determine the infection status by counting oocyst numbers. At ~21–23 days post-feeding, three or five mosquitoes were randomly selected from the cage and allowed to feed on noninfected mice (challenge). Mosquitoes that did not take a blood meal were replaced until the final number of mosquitoes for each group (three or five) was reached. The salivary glands of most mosquitoes were dissected for counting sporozoites. A total of five mice were used per experiment and three biological replicates were conducted for a total of 15 mice per mosquito group. After mosquito challenge, mice were monitored daily for 14 days to determine blood stage infection using Giemsa-stained blood smears.

## Detection of MP2 and scorpine protein ingestion

Ingestion of salivary gland-expressed MP2 and scorpine into the midgut lumen was analyzed with a low-melting agarose feeding assay (*Oliveira et al., 2011*). WT and Sg/E females were fed with a 1% low-melting agarose solution for 15 min. One half hour later, midguts were collected from five mosquitoes on ice-cold PBS. The midgut agarose bolus was placed in 1% protease inhibitor cocktail (Sigma, Cat# P8340-1 ML) and 0.1 mM PMSF, homogenized, and stored at −80°C. Proteins in the homogenate were resolved by SDS-PAGE and transferred to a PVDF membrane. The membrane was probed with a mouse anti-MP2 and scorpine antibodies (1:2000) and developed as described before. An antibody against the saliva protein AAPP (*Hayashi et al., 2012*) was used as a positive saliva control (1:2000).

## Acknowledgements

We thank the Insectary and Parasite Core Facilities of the Johns Hopkins Malaria Research Institute. This work was supported by a grant R01AI031478 from the National Institutes of Health and by the Bloomberg Philanthropies. Supply of human blood was supported by the National Institutes of Health grant RR00052. We thank Dr. Yuemei Dong from the Johns Hopkins Malaria Research Institute for providing the helper plasmid and Dr. Christopher Potter from Johns Hopkins School of Medicine for providing pXL-BACIIECFP-15XQUAS-TATA-PAI-SV40 and pXL-BACII-DsRed-AAPP-QF2-hsp70 plasmids.

## Additional information

### Funding

| Funder | Grant reference number | Author |
| --- | --- | --- |
| National Institutes of Health | R01AI031478 | Marcelo Jacobs-Lorena |
| Bloomberg Philanthropies | | Marcelo Jacobs-Lorena |

The funders had no role in study design, data collection and interpretation, or the decision to submit the work for publication.

### Author contributions

Wei Huang, Conceptualization, Data curation, Formal analysis, Validation, Visualization, Methodology, Writing - original draft; Joel Vega-Rodriguez, Validation, Investigation, Writing – review and editing; Chritopher Kizito, Investigation, Methodology; Sung-Jae Cha, Formal analysis, Investigation, Writing – review and editing; Marcelo Jacobs-Lorena, Conceptualization, Resources, Formal analysis, Supervision, Funding acquisition, Project administration, Writing – review and editing

### Author ORCIDs

Wei Huang http://orcid.org/0000-0003-0300-7470
Joel Vega-Rodriguez http://orcid.org/0000-0002-9576-0058
Marcelo Jacobs-Lorena http://orcid.org/0000-0003-0449-432X

Decision letter and Author response
Decision letter https://doi.org/10.7554/eLife.77584.sa1
Author response https://doi.org/10.7554/eLife.77584.sa2

## Additional files

### Supplementary files

• Appendix 1—figure 1—source data 1. 'Appendix 1-Figure 1-DNA gel of PCR verification 1.tif and Appendix 1-Figure 1-DNA gel of PCR verification 2.tif' are the original DNA gel for *Appendix 1— figure 1*.

• Appendix 1—figure 2—source data 1. Source data of *Appendix 1—figure 2A, B, C, D, E and F*. 'Appendix 1-Figure 2-source data.xlsx' is the original data for *Appendix 1—figure 2A–E*; 'Appendix 1-Figure 2-source data.pzf' shows that *Appendix 1—figure 2A–F* were generated with GraphPad Prism; 'Appendix 1-Figure 2-source data-Female Survival Curve P value.pzf' and 'Appendix 1-Figure 2-source data-Male survival curve P value.pzf' shows the calculation of p-value with GraphPad Prism.

• Appendix 1—figure 3—source data 1. Source data of *Appendix 1—figure 3B*. 'Appendix 1-Figure 3A and B-source data-life span.pzf' shows that *Appendix 1—figure 3A and B* were generated with GraphPad Prism; 'Appendix 1-Figure 3-source data-Female Survival Curve P value.pzf' and 'Appendix 1-Figure 3-source data-Male Survival Curve P value.pzf' show the calculation of p-value for *Appendix 1—figure 3A and B* with GraphPad Prism, respectively; 'Appendix 1-Figure 3C and D-source data-fecundity and fertility.pzf' was generated with GraphPad Prism.

• Appendix 1—figure 4—source data 1. 'Appendix 1-Figure 4B-source data-blood feeding in feeder. xlsx' is the original colony-forming unit (CFU) data for *Appendix 1—figure 4B*; 'Appendix 1-Figure 4B-source data-blood feeding in feeder.pzf' shows that *Appendix 1—figure 4B* was generated with GraphPad Prism.

• Appendix 1—figure 5—source data 1. 'Appendix 1-Figure 5-source data.pzf' shows that *Appendix 1—figure 5* was generated with GraphPad Prism.

• Appendix 1—figure 6—source data 1. Pictures of 'Appendix 1-Figure 6-western blot-α-tubulin in midgut.tif," 'Appendix 1-Figure 6-western blot-multi-effectors (Scorpine antibody) in midgut.tif,' 'Appendix 1-Figure 6-western blot-scoprine in midgut.tif,' and 'Appendix 1-Figure 6-western blot-MP2 in midgut.tif' are original images of Western blots detected with rabbit anti-α-tubulin antibody, mouse anti-scorpine, mouse anti-scorpine, and mouse anti-MP2.

• Appendix 1—figure 7—source data 1. Pictures of 'Appendix 1-Figure 7-western blot-AAPP in midgut by digestion.tif,' 'Appendix 1-Figure 7-western blot-MP2 in midgut by digestion.tif,' and 'Appendix 1-Figure 7-western blot-Scorpine in midgut by digestion.tif' are original images of Western blots detected with mouse anti-AAPP, mouse anti-MP2, and mouse anti-scorpine.

• Transparent reporting form

• Source data 1. Source data of *Figures 1–4*, *Appendix 1—figures 1–7*.

### Data availability

All data generated or analysed during this study are included in the manuscript and supporting file.

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

## Appendix 1

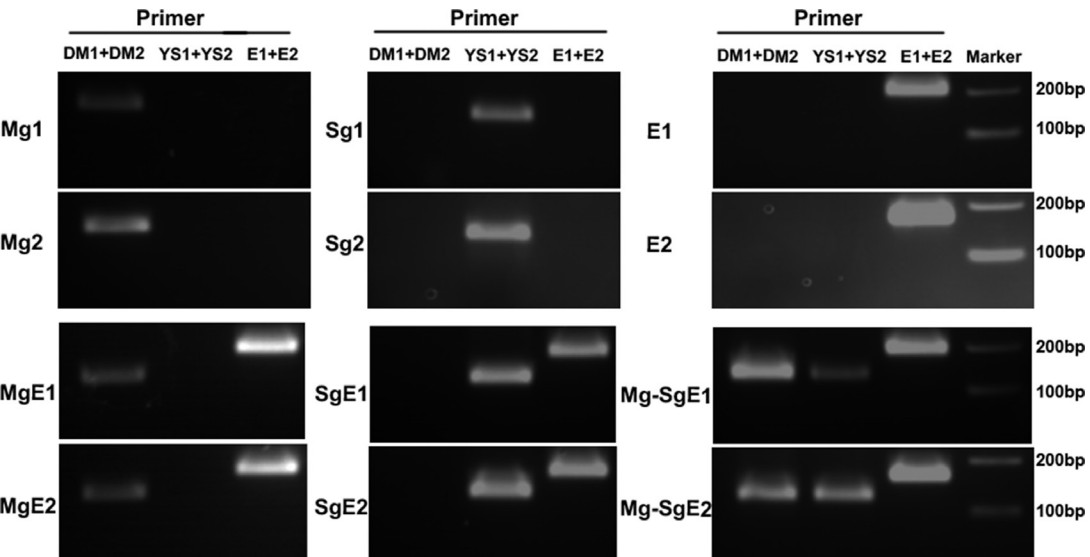

**Appendix 1—figure 1.** PCR validation of plasmid insertion in mosquito lines. Primer pairs used for PCR reactions are indicated on top of each lane (sequences provided in *Appendix 1—table 5*, *Appendix 1—table 6*, and *Appendix 1—table 7*; position of primers indicated in *Figure 1A* with red font). The DM1 + DM2 primer pair was used to verify the MG QF2 driver plasmid insertion; the YS1 + YS2 primer pair was used to verify the SG QF2 driver plasmid insertion; and the E1 + E2 primer pair was used to verify the QUAS-MP2-QUAS-scorpine effector plasmid insertion. The transgenic lines are identified to the left of each panel.

The online version of this article includes the following source data for appendix 1—figure 1:

**Appendix 1—figure 1—source data 1.** 'Appendix 1-Figure 1-DNA gel of PCR verification 1.tif and Appendix 1-Figure 1-DNA gel of PCR verification 2.tif' are the original DNA gel for *Appendix 1—figure 1*.

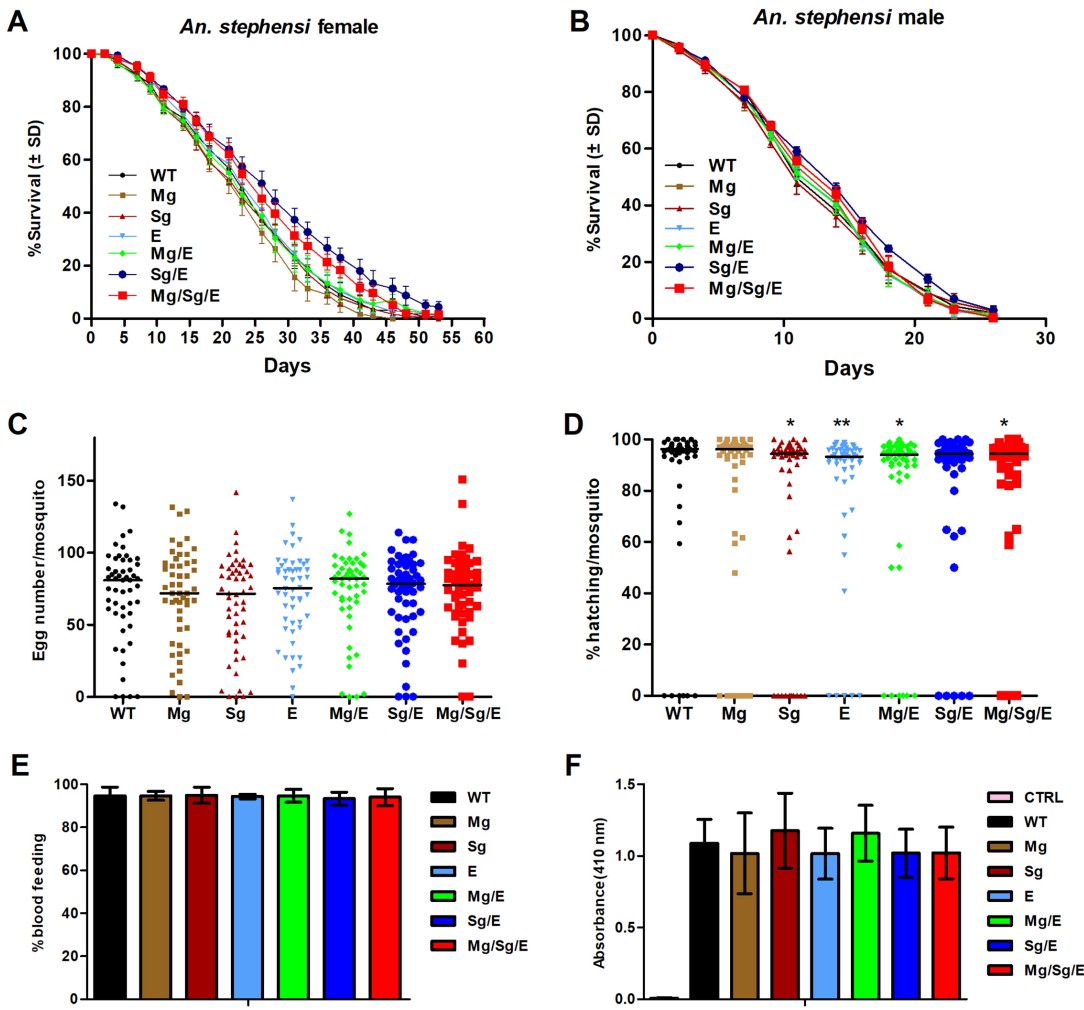

**Appendix 1—figure 2.** Fitness analysis of *An. stephensi* transgenic lines. (**A, B**) Survival curves for wild-type (WT) and transgenic (see *Figure 1A*) females that received one blood meal on day 2 (**A**) and males (**B**), all maintained on sugar meal. No significant differences in survival rate were detected as calculated by Kaplan–Meier survival curves, and multiple comparisons by log-rank test with Bonferroni correction for parental expressing lines [(**A**) WT and Mg: p=0.9855; WT and Sg: p=0.6524; WT and E: p=0.4602; WT and Mg/E: p=0.6417; WT and Sg/E: p=0.3237; WT and Mg/Sg/E: p=0.3712; (**B**) WT and Mg: p=0.6643; WT and Sg: p=0.5283; WT and E: p=0.9875; WT and Mg/E: p=0.9650; WT and Sg/E: p=0.0753; WT and Mg/Sg/E: p=0.3387]. Combined from three biological replicates (N: 300 mosquitoes). (**C**) Comparison of fecundity (number of laid eggs) between WT and transgenic mosquitoes. No significant differences were found using log-rank (Mantel–Cox) test. (**D**) Comparison of fertility (proportion of laid eggs that hatched) between WT and transgenic lines. Statistical analysis was done by Mann–Whitney *U*-test. [(**C**) WT and Mg: p=0.7426; WT and Sg: p=0.2379; WT and E: p=0.7748; WT and Mg/E: p=0.6988; WT and Sg/E: p=0.8048; WT and Mg/Sg/E: p=0.9663; (**D**) WT and Mg: p=0.9325; *WT and Sg: p=0.0211; **WT and E: p=0.0089; *WT and Mg/E: p=0.0315; WT and Sg/E: p=0.0502; *WT and Mg/Sg/E: p=0.0324]. (**C, D**) Data combined from three biological replicates (N: 60 mosquitoes); horizontal lines are median values. (**E**) The percentage of mosquitoes that take a blood meal is not affected. Two-day-old female mosquitoes were allowed to feed on mice, and the percentage of mosquitoes that fed was determined after 30 min feeding. (**F**) Amount of blood uptake is not affected. Quantification of protein-bound heme at 410 nm from midguts of WT and transgenic mosquitoes before (CTRL) and after a blood meal. (**E, F**) Error bars represent SD of the mean; data pooled from three independent experiments; no significant differences were found using Student's *t*-test [(**E**) WT and Mg: p=0.9027; WT and Sg: p=0.9027; WT and E: p=0.9216; WT and Mg/E: p=0.9001; WT and Sg/E: p=1; WT and Mg/Sg/E: p=0.6779; (**F**) WT and Mg: p=0.7274; WT and Sg: p=0.6497; WT and E: p=0.6321; WT and Mg/E: p=0.6642; WT and Sg/E: p=0.6351; WT and Mg/Sg/E: p=0.7570].

The online version of this article includes the following source data for appendix 1—figure 2:

**Appendix 1—figure 2—source data 1.** Source data of *Appendix 1—figure 2A, B, C, D, E and F*.

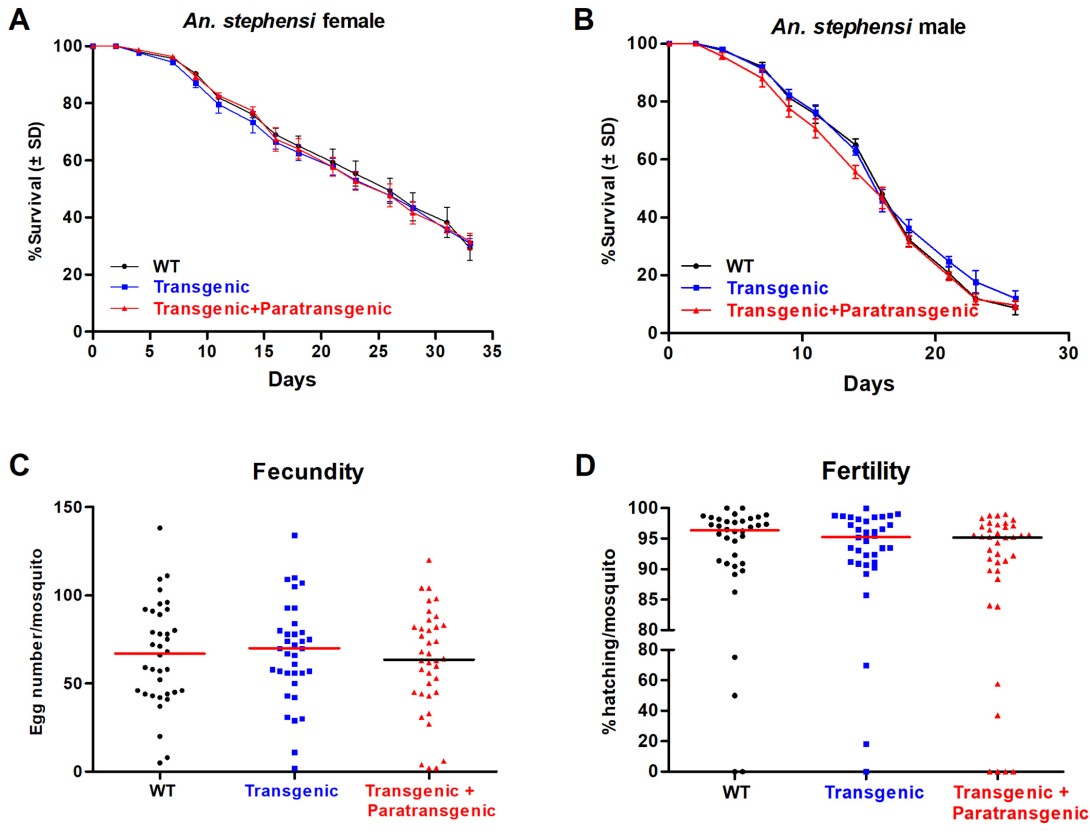

**Appendix 1—figure 3.** Fitness analysis of *An. stephensi* transgenic lines in combination or not with paratransgenesis. (**A, B**) Survival curves for wild-type (WT) and transgenic (see *Figure 1A*) females in combination or not with paratransgenesis that received one blood meal on day 2 (**A**), and males (**B**), all maintained on sugar meal. No significant differences in survival rate were detected as calculated by Kaplan–Meier survival curves, and multiple comparisons by log-rank test with Bonferroni correction for parental expressing lines [(**A**) WT and transgenic: p=0.1650; WT and (transgenic + paratransgenic): p=0.3362; transgenic and (transgenic + paratransgenic): p=0.6499; (**B**) WT and transgenic: p=0.7587; WT and (transgenic + paratransgenic): p=0.0912; transgenic and (transgenic + paratransgenic): p=0.0759]. Data pooled from three biological replicates (N: 300 mosquitoes). (**C**) Comparison of fecundity (number of laid eggs) between WT, transgenic mosquitoes, and (transgenic + paratransgenic) mosquitoes. No significant differences were found using log-rank (Mantel–Cox) test. (**D**) Comparison of fertility (proportion of laid eggs that hatched) between WT, transgenic lines, and (transgenic + paratransgenic) mosquitoes. Statistical analysis was done by Mann–Whitney *U*-test. [(**C**) WT and transgenic: p=0.9039; WT and (transgenic + paratransgenic): p=0.7661; transgenic and (transgenic + paratransgenic): p=0.6587; (**D**) WT and transgenic: p=0.7085; WT and (transgenic + paratransgenic): p=0.1024; transgenic and (transgenic + paratransgenic): p=0.2766]. (**C, D**) Data pooled from three biological replicates (N: 45 mosquitoes); horizontal lines are median values.

The online version of this article includes the following source data for appendix 1—figure 3:

**Appendix 1—figure 3—source data 1.** Source data of *Appendix 1—figure 3B*.

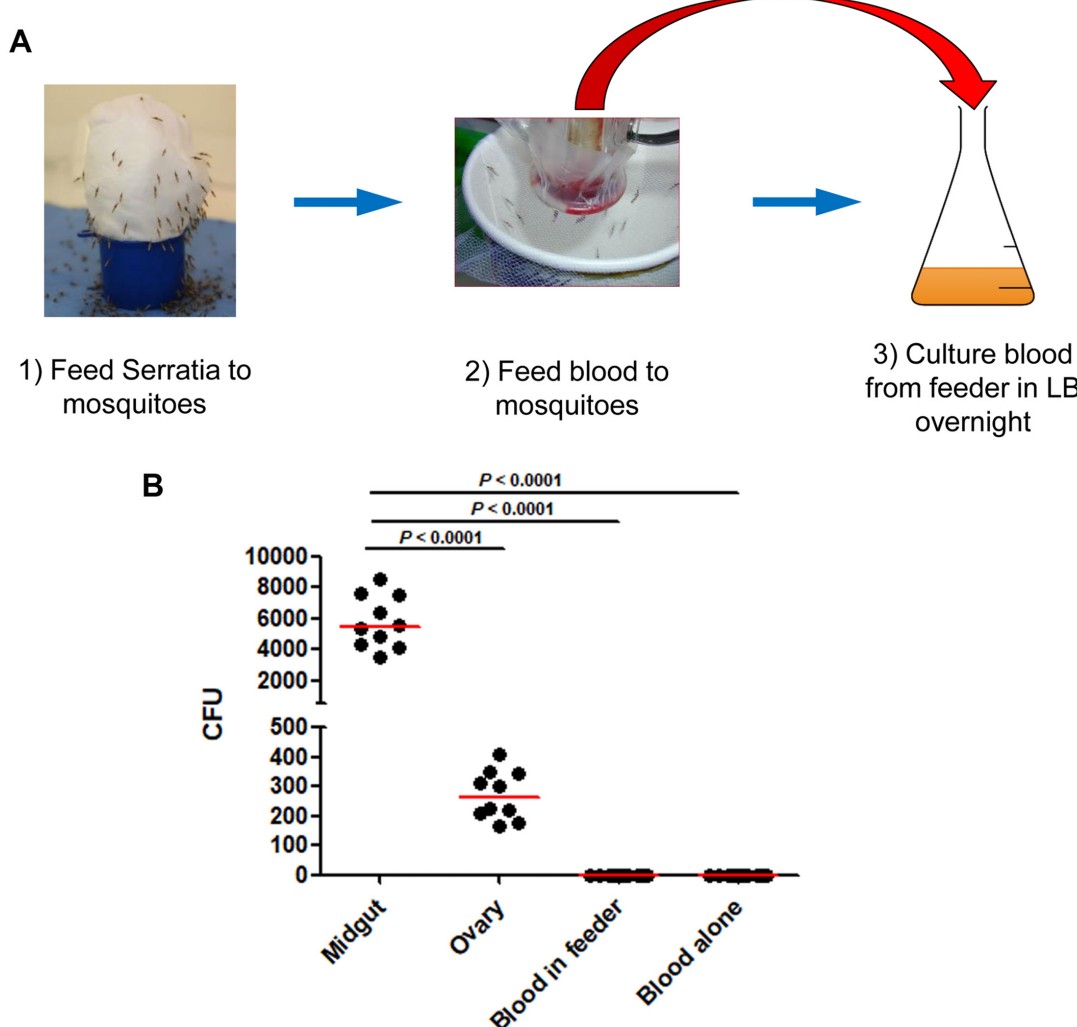

**A**

1) Feed Serratia to mosquitoes

2) Feed blood to mosquitoes

3) Culture blood from feeder in LB overnight

**Appendix 1—figure 4.** Assay of bacteria released by mosquitoes during feeding. (**A**) Two-day-old mosquitoes were fed overnight on $10^7$ *Serratia*-GFP/ml of 5% sugar plus food dye. Mosquitoes carrying the food dye marker were maintained on sterile 5% sugar for 2 days. At this point, 100 female mosquitoes were starved for 3 hr and fed blood for 30 min using a membrane feeder. To estimate the number of *Serratia*-GFP bacteria in the midgut and ovary, an additional 10 female mosquitoes were dissected prior to blood feeding, and dilutions of the homogenates were plated on LB/kanamycin plates. After blood feeding, 50 µl of blood from the feeder (out of 300 µl initial volume) was added to 5 ml LB/kanamycin, grown overnight, and then plated on LB/kanamycin plates to detect presence (or not) of *Serratia*-GFP in the blood. (**B**) Bacteria numbers in midguts, ovaries, and blood from the feeder. No bacteria were detected from the blood samples. Data pooled from 10 independent biological experiments. Statistical analysis was done by Mann–Whitney *U*-test.

The online version of this article includes the following source data for appendix 1—figure 4:

**Appendix 1—figure 4—source data 1.** 'Appendix 1-Figure 4B-source data-blood feeding in feeder.xlsx' is the original colony-forming unit (CFU) data for ***Appendix 1—figure 4B***; 'Appendix 1-Figure 4B-source data-blood feeding in feeder.pzf' shows that ***Appendix 1—figure 4B*** was generated with GraphPad Prism.

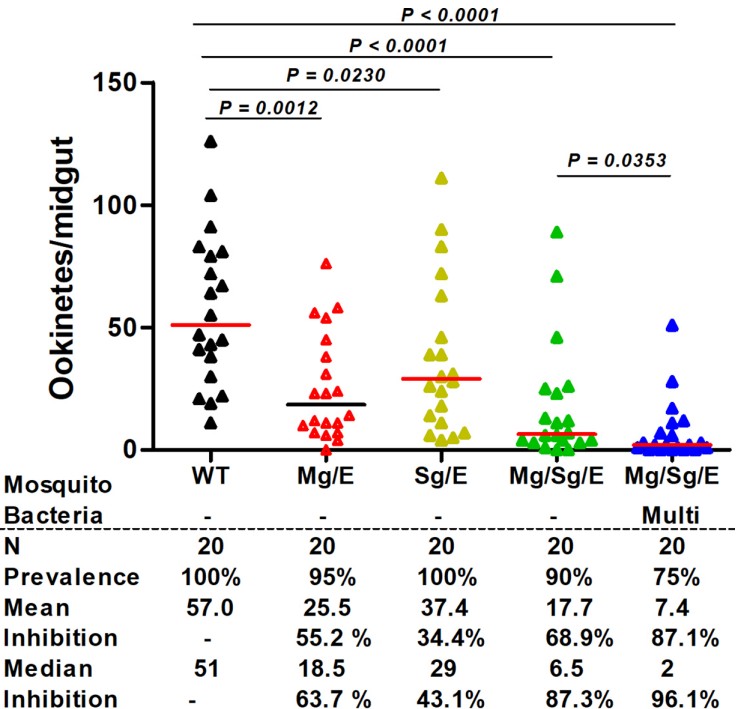

**Appendix 1—figure 5.** Transgenesis and paratransgenesis strongly impair *Plasmodium* development in the mosquito midgut. Two-day-old *An. stephensi* mosquitoes were fed (or not) overnight with wild-type or recombinant *Serratia* AS1-multi bacteria, as indicated. After 48 hr, all mosquito groups were fed on the same *P. falciparum* gametocyte culture and midgut ookinete numbers were determined at 22 hr. Data from one experiment. Statistical analysis was done by Mann–Whitney *U*-test. 'Multi': *Serratia* AS1-multi bacteria expressing multiple effectors; N: number of mosquitoes assayed; Prevalence: proportion of mosquitoes carrying one or more parasite.

The online version of this article includes the following source data for appendix 1—figure 5:

**Appendix 1—figure 5—source data 1.** 'Appendix 1-Figure 5-source data.pzf' shows that *Appendix 1—figure 5* was generated with GraphPad Prism.

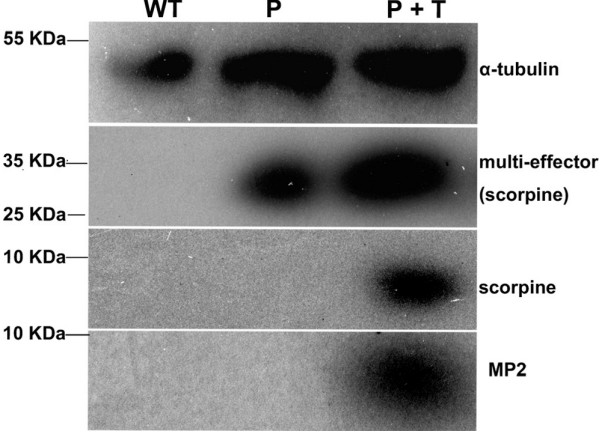

**Appendix 1—figure 6.** Immunoblotting showing the mosquito-expressed multiple-effector protein (detected with a scorpine antibody), and the bacteria-expressed MP2 peptide and scorpine, in midguts collected 1 day after an infected blood meal. P: paratransgenesis; P+T: combination of paratransgenesis and transgenesis. Tubulin served as a loading control.

The online version of this article includes the following source data for appendix 1—figure 6:

**Appendix 1—figure 6—source data 1.** Pictures of 'Appendix 1-Figure 6-western blot-α-tubulin in midgut.tif," 'Appendix 1-Figure 6-western blot-multi-effectors (Scorpine antibody) in midgut.tif,' 'Appendix 1-Figure 6-western blot-scoprine in midgut.tif,' and 'Appendix 1-Figure 6-western blot-MP2 in midgut.tif' are original images of Western blots detected with rabbit anti-α-tubulin antibody, mouse anti-scorpine, mouse anti-scorpine, and mouse anti-MP2.

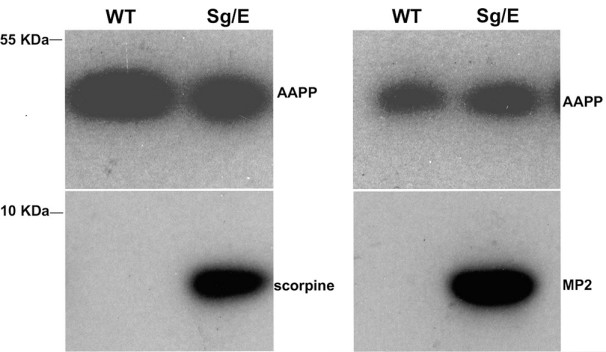

**Appendix 1—figure 7.** Immunoblotting showing effector ingestion together with saliva during the probe. WT and Sg/E female mosquitoes were fed with low melting agarose solution, and then MP2 and Scorpine peptides were detected in the mosquito midgut.

The online version of this article includes the following source data for appendix 1—figure 7:

**Appendix 1—figure 7—source data 1.** Pictures of 'Appendix 1-Figure 7-western blot-AAPP in midgut by digestion.tif,' 'Appendix 1-Figure 7-western blot-MP2 in midgut by digestion.tif,' and 'Appendix 1-Figure 7-western blot-Scorpine in midgut by digestion.tif' are original images of Western blots detected with mouse anti-AAPP, mouse anti-MP2, and mouse anti-scorpine.

**Effectors sequence for QUAS-E plasmid**

ATGGTGCGATTAAACAGTGCAGCCGGCTCCCGGTGGTGGGCCCCAGCGATGGCCATCCTGGC
GGTGGCGCTCAGTGTCGAAGCGGCCTGCTACATCAAGACCCTGCACCCCCCCTGCTCCGGCT
CCCCGGCGCCTGCTACATCAAGACCCTGCACCCCCCCTGCTCCGGCTCCCCCGGCGCCTGC
TACATCAAGACCCTGCACCCCCCCTGCTCCGGCTCCCCCGGCGCCTGCTACATCAAGACCCTG
CACCCCCCCTGCTCCTAAACGTCGATCTTTGTGAAGGAACCTTACTTCTGTGGTGTGA
CATAATTGGACAAACTACCTACAGAGATTTAAAGCTCGGGTAATCGCTTATCCTCGGATA
AACAATTATCCTCACGGGTAATCGCTTATCCGCTCGGGTAATCGCTTATCCTCGGGTAATCGCTTA
TCCTTGGGTAATCGCTTATCCTCGGATAAACAATTATCCTCACGGGTAATCGCTTATCGCTCGGG
TAATCGCTTATCCTCGGGTAATCGCTTATCCTTGGGTAATCGCTTATCCTCGGATAAACAATTATCC
TCACGGGTAATCGCTTATCCGCTCGGGTAATCGCTTATCCTCGGGTAATCGCTTATCCTTTCACGT
TGGGACTCAGTGAGGAGGACCTGAATTCCTGCAGCCCGAGCGGAGACTCTAGCGAGCGCCGG
AGTATAAATAGAGGCGCTTCGTCTACGGAGCGACAATTCAATTCAAACAAGCAAAGTGAACACGT
CGCTAAGCGAAAGCTAAGCAAATAAACAAGCGCAGCTGAACAAGCTAAACAATCTGCAGTAAAGT
GCAAGTTAAAGTGAATCAATTAAAAGTAACCAGCAACCAAGTAAATCAACTGCAACTACTGAAATC
TGCCAAGAAGTAATTATTGAATACAAGAAGAACTCTGAATAGGGAATTGGGATGGTGCGATTAA
ACAGTGCAGCCGGCTCCCGGTGGTGGGCCCCAGCGATGGCCATCCTGGCGGTGGCGCTCAGT
GTCGAAGCGGGTTGGATCAATGAGGAAAAAATCCAAAAGAAAATCGATGAGCGCATGGGTAATA
CCGTGCTCGGCCGTATGGCCAAGGCCATCGTCCACAAGATGGCTAAAAATGAGTTCCAATGTAT
GGCCAATATGGACATGCTCGGTAACTGTGAAAAGCATTGCCAAACCTCCGGTGAGAAGGGTTAT
TGCCATGGTACTAAATGCAAGTGTGGTACTCCACTCTCCTACTAACTCGAGACGTCGATCT

**YFP**

ATGGTGAGCAAGGGCGAGGAGCTGTTCACCGGGGTGGTGCCCATCCTGGTCGAGCTGGACGG
CGACGTAAACGGCCACAAGTTCAGCGTGTCCGGCGAGGGCGAGGGCGATGCCACCTACGGCA
AGCTTTCGCGTCGGTAGTAGGACCCTGAAGTTCATCTGCACCACCGGCAAGCTGCCCGTGCCC
TGGCCCACCCTCGTGACCACCTTCGGCTACGGCGTGCAGTGCTTCGCCCGCTACCCCGACCAC
ATGCGCCAGCACGACTTCTTCAAGTCCGCCATGCCCGAAGGCTACGTCCAGGAGCGCACCATC
TTCTTCAAGGACGACGGCAACTACAAGACCCGCGCCGAGGTGAAGTTCGAGGGCGACACCCT
GGTGAACCGCATCGAGCTGAAGGGCATCGACTTCAAGGAGGACGGCAACATCCTGGGGCACAA
GCTGGAGTACAACTACAACAGCCACAACGTCTATATCATGGCCGACAAGCAGAAGAACGGCATC
AAGGTGAACTTCAAGATCCGCCACAACATCGAGGACGGCAGCGTGCAGCTCGCCGACCACTAC
CAGCAGAACCACCCCCATCGGCGACGGCCCCGTGCTGCTGCCCGACCAACCACTACCTGAGCTAC
CAGTCCGCCCTGAGCAAAGACCCCAACGAGAAGCGCGATCACATGGTCCTGCTGGAGTTCGTG
ACCGCCGCCGGGATCACTCTCGGCATGGACGAGCTGTACAAGTAG

**AsAAPP (anopheline antiplatelet protein gene) promoter**

GGACTTCGCGTCGGTAGTAGTATTCTCCGGCAACGCTTTCCCAACCGTGATCGCGAAGTCCCTC
GCCACAACAGCTTGCCTCAGCCGATCCGTATTGAGCCTAGAAGTAGGCTGATAGCGCTGCGTAT
TGGCTACGCAGAGCTTTTGGCGTAACTTTACCATAACCAGGACATGAGTTGACTTTTGCTCCTCT
GTAGGTCCGTAGGCCGGTTATATCCGAGAAGTGCCTCCCATCGATGAGAACGAGGTCAATCTGG
GAATATGTCTGCTGTGGTGATCTCCAGGTGTAGCTGAACCGAGGTGCATCATCAAGAGCTCATT
GTCGTTCGTCAGCGGGTGGGTGCTGAAACTTCCTATTGTGGGTTTGTATGCCTCCTCCCGTCCG
ACCTGAGCGTTAAAATCTCCGATGACTATCTTCACATCCTGTTGTGGGCAGCAGATGTGCTCCAT
CTCCAACTGCGTATAGAGGATCTCCTTCTCTTCTTCGGGGCTTCCAAGGTGCGGAATGTACACAT
TAATAATGCTGAGGTTGAAGAACCTGCCACGGATCCTCAGCCTGCACCGTAAATGATCTGCAGTT
CCATGTCTCGAGTTTCCGATCGTGTGTTTTGTTTCGTAGCGTTGGTCTACACGTTGAATCTGATT
CGAAATTTCATGGTTTTCGTTCGTTGCATTTATATTGAGGGTGGCTTGCAAGGCCTCTCCTCCGA
CATCTTATCTCGTCGGAGGGCCTACGCTTTTTAGCTGTTTAAAGTCCCGTAGGACAAGAGAGACG
GCCAGCCGCCCCTAACATGCAGAACAGAGGCTAGCTATCCGCCCCCCCCCCCCCCCCTCAATTTA
GCCATATCACCCATCTTCCCATGGGATTCGGTTTCCCCGTATACCCAAGCTCAGATTTTTGGAGA
GACATACCATTCTGTATGCCGGGGACGTATTGTGTCTAAGTGGGGTGTGGAGAGCCTATGTTAAC
CTTGAAGAGGCTTCGGATCCATACCCTACTATATACAATATGGGAAGATGGTTTATATATTGGGAAAA
ATCCATCTGAGCATCTCCCTCTCAAACGCGGCTAAGAAGGTCAAAGCTAAGATGTCTCTAAGGC
GTATGTGTGTGACTGGGACTATACATGTTCTGTACAGTCCCAGCATCGTCCGCGGCGGACAAGT
ATTTTGAGTGGAGAACTTTTCTCATGACCAGTTGACAGCCAGCACAACTCAACATCAATGTTGGT
GTCGGTACTGATTTTTGACCCCTGATAAGTGAAGTTTTGAACGATTTCAAAGATACGATCACCTCG
CCAAAGTTGGTGGTTTTCGTCCCCTTTAACTTCAACCCGAGGCTAAGTGCCGATCGTTCGATCCT
TTGGAAAGCTAGGATAGAAGAGCCTCAAACCAATGATGTCTATGTCATTTATGTTTGGTGCGGATA
CACCGCCAGTGAAACAAGTGTAATAAGAGTAAAATTTGATTATTTAATATTTATTTACCTTCGCATAA
TTTATTTGACGAACTTAGCATGATTTATATCTGTTGGCAAAATGAGACAATTTAATCATGTTCTGCAT
TTACTTTAATGATCATCGTATTGCAACATGCTTGTTTAACCATTATTTATCAAACATTACTTTTCCCTC
TACTTGTGCACAGTAAAGCAGGTGCCATTTGCTTAACACTAACAAAAGCTGTACAAATACTAGGG
ACGGCTCCGAGGTCCGACGATCGTCCGAGTTTTCCCTTATAAAAGCGCCGCGATCGTAACGTAA
AGATCATCAGTAGAGTTTGCCTTTTTTTCCATAGCTCACAGGTGAATAAACG

**AsAper1 (adult peritrophic matrix protein 1 gene) promoter**

TACCGGCAATACTGGTTGTTGAGGGGAAAAAATAGACAATAAAATTTTTCTTTCAGCTTTGAATGA
TGGATAATGCTTTATTTCGGTTCAGGAGCTTGTATTGCTTCGCTGGATAACTGGAACTTGGAATCA
ACGTTGTACAGGTTGTAAATGTTCCTCAAGCCCTGGAAAAAGTGTCTACATCTTCAAAACGATGA
GCGATTTTGATATAAAGACAACAGTTTTGGGGGTGACCGGAGCGATAACTTTCTTTAACTTTAACT
GAGTTCTTTCGAGATTATAGAACCAAAGAAGAAGATATCAATTTTGATATAATCGTACATGCTTTTAA
ACATAAAGACACTTTCTTTATGATCTAAGAAACACTGTACACACACTTATTACATGCCGTCCAATAT
TGTTCAGTGCTTAAGTATTGAGGTTAATTGTTAATGTTAGAATAAAGCCTTCTACACTACACGGCAA
GGAGTGTAAAGCAGTGATGATGATGATGGTGATTATTCGGATTATCAAGTTGATAAAGAATTAGCT
GTAATGTTTTAACACCTCCACCGTACATTGTCCACTCCTTCTCCGATCCAGCTCCGTTCGGTGTG
ACATCAATTGGTTACAGCAGAATAATCACTTTATACCAATATGATGTCACCATCAAGCGTATCGACA
CAAACTGTCTGTCTCACTCTCTTGTGTTCTACCAATTTGGGGACCAATGCCATAGGTGTCCTGCC
CCGGGGTGTGTCTGCCCCCTGACCGGGACACATGTCCGTTGCGAGTGTTTTATAAACGCGCGC
ACACCCAATAACCATAATCGTAACGATGAACGGCTTGTGTGCCTATCGGAAAAGCGTCTGCTTTT
TTTGGGAGGGCTGCTTCAATTGATGCAGTTAAAAAGGGAACGAGCGTGATGATTATGATTACAAC
TAATATGCATATGAGTGGAGCCGTGTGGAGGATAATCCACGCCACACTCTGGTGGATGCGTCCG
ATGGAGACACTTTAAATAAATGGCCAAATTTGCAGTTCATAAAATCGAATGAAGGATAACCAGAAT
GAACGTTAGGGAAGGTATGAGTTGGAGTAGGAAGTCAAAATTTAAACCAAAAAAAGTACGCGGGG
GATGGTATTGGCAAGAGGTCCAACTTCAAGGGGTCAAGGGGCATTTAGAAGCTTATGTACTTCTC
CAAAGTTAGTCATCAGCAGACAGCTTTAGTAGATCGCCAAATAGACACGAGCACCGTGCTGAGAT
TCACTTTTGATAGCCGATAGTCGATAAGTTGAAAGTGTCCCACCATCCAAGCAGCGTGCCAGTTA
AATCAACTGTACACCGTTTCGAGAGGCAGCGATGGACAGTGCTTAAAATGTTGCTAACAGTCAGT
CCACTTCAGATTATCCATGCTATCGGAACCGTAGCGCCTCCAATAAAAAGGACACACCGGACGGT
GCAACAGGTACAGTACGTAACGTGCGTTGACGGTAATACATACAACTCGCGGCATCTAACATTCT
CATC

**Appendix 1—figure 8.** Sequences of the synthetic transgenes on the plasmid constructs for the transformation of *Anopheles* mosquito embryos.

**Appendix 1—table 1.** Transgene integration sites.

| Line | Integration site | Integration in gene |
|------|------------------|---------------------|
| Mg1 | AsteS1:KB664810:1:1229869:1 | No |
| Mg2 | AsteS1:KB664721:1:1159608:1 | No |
| Sg1 | AsteS1:KB664422.1 | No |
| | AsteS1:KB664506.1 | No |
| | AsteS1: KB664514.1 | Gamma-glutamyltranspeptidase (ASTE010947) |
| Sg2 | AsteS1: KB664921.1 | No |
| E1 | AsteS1:KB664810:1:1229869:1 | No |
| | AsteS1:KB664810:1:1229869:1 | No |
| E2 | AsteS1:KB664538:1:382792:1 | No |

Integration site: AsteS1; contig number: precision site.

**Appendix 1—table 2.** Verification of transgene homozygosity.

| Mosquito line | Larva numbers | Red | Yellow | Blue |
|---------------|---------------|-----|--------|------|
| **Mg1** | **412** | **412** | | |
| Mg2 | 276 | 276 | | |
| Sg1 | 178 | | 178 | |
| Sg2 | 329 | | 329 | |
| E1 | 206 | | 206 | |
| E2 | 378 | | 378 | |
| Mg/E1 | 262 | 262 | | 262 |
| Mg/E2 | 198 | 198 | | 198 |
| Sg/E1 | 307 | | 307 | 307 |
| Sg/E2 | 345 | | 345 | 345 |
| Mg/Sg/E1 | 228 | 228 | 228 | 228 |
| Mg/Sg/E2 | 361 | 361 | 361 | 361 |

A total of 20 transgenic female mosquitoes from each line were mated with wild-type males and the progeny larvae assayed for expression of the dominant eye fluorescence marker. No non-fluorescent larvae were found, indicating that the females were homozygous for the transgenes.

**Appendix 1—table 3.** Expression of MP2 and scorpine mRNAs relative to the endogenous AsAper mRNA, quantified by qRT-PCR in the midgut of transgenic mosquitoes.

| Mosquito lines | Relative expression in midgut | | |
|----------------|--------|----------|------|
| | AsAper | Scorpine | MP2 |
| E | 1.0 ± 0.2 | N | N |
| Mg/E | 1.0 ± 0.3 | 44.3 ± 10.7 | 49.3 ± 16.7 |
| Mg/Sg/E | 1.0 ± 0.2 | 56.1 ± 8.7 | 35.6 ± 8.9 |

*Appendix 1—table 3 Continued on next page*

*Appendix 1—table 3 Continued*

| Mosquito lines | Relative expression in midgut | | |
|---|---|---|---|
| | AsAper | Scorpine | MP2 |

The rpS7 gene was used as reference, and WT mosquitoes were used as negative controls. Identification of mosquito lines provided in **Figure 1A**. N: transcript not detected. Data pooled from three independent biological experiments. Mean ± SD.

**Appendix 1—table 4.** Relative expression of MP2 and scorpine mRNAs relative to the endogenous AsAAPP mRNA quantified by qRT-PCR in the salivary glands of transgenic mosquitoes.

| Mosquito lines | Relative expression in salivary gland | | |
|---|---|---|---|
| | AsAAPP | Scorpine | MP2 |
| E | 1.0 ± 0.2 | N | N |
| Sg/E | 1.0 ± 0.1 | 27.3 ± 15.6 | 49.1 ± 7.6 |
| Mg/Sg/E | 1.0 ± 0.3 | 62.5 ± 9.3 | 140.2 ± 38.3 |

The rpS7 gene was used as reference, and WT mosquitoes were used as negative controls. Identification of mosquito lines provided in **Figure 1A**. Data pooled from three independent experiments. Mean ± SD.

N: transcript not detected.

**Appendix 1—table 5.** *Serratia* is horizontally (sexually) transmitted.

| Males carrying AS1-multi | Females (mated/virgin) | Female CFUs | | |
|---|---|---|---|---|
| | | Spermatheca | Midgut | Ovary |
| WT | WT mated | 0 | 0 | 0 |
| Transgenic | Transgenic mated | 0 | 0 | 0 |
| WT | WT virgin | 3.9 ± 4.7 | 115 ± 118 | 11 ± 7.7 |
| Transgenic | Transgenic virgin | 2.9 ± 4.7 | 104 ± 111 | 8.7 ± 7.8 |

Newly emerged virgin male adult mosquitoes were fed overnight on 5% sugar solution containing $10^7$ *AS1*-multi bacteria/ml and placed with females. Three days later, 10 females were assayed for the presence of *Serratia AS1* by plating midgut, ovary, and spermatheca homogenates on apramycin/ampicillin agar plates and colonies were counted. Transgenic mosquito: Mg/Sg/E. Mated females were used as controls as female mosquitoes mate only once in their lifetimes. Data pooled from three independent experiments. Mean ± SD.

CFU: colony-forming unit.

**Appendix 1—table 6.** Vectors used in this research.

| Vectors | Reference/notes |
|---|---|
| phsp-pBac | (**Handler and Harrell, 1999**) |
| pXL-BACII-DsRed-AAPP-QF2-hsp70 | (**Potter et al., 2010b**) |
| pXL-BACIIECFP-15XQUAS-TATA-PAI-SV40 | **Potter et al., 2010b** |
| pXL-BACII- DsRed-Aper-QF2-Hsp70 | Mg QF2 driver plasmid |
| pXL-BAC-YFP-AAPP promoter-QF2-Hsp70 | Sg QF2 driver plasmid 2575 |
| pXL-BACIIECFP-15XQUAS-TATA-MP2-SV40-15XQUAS-TATA-Scorpine-SV40 | QUAS-MP2-Scorpine effector plasmid |
| pBAM2-YFP | DNA template for YFP |

**Appendix 1—table 7.** Oligonucleotide primers used in this study.

| Primer | Sequence (5′–3′) | Notes |
|---|---|---|
| MgPF | ATCAATGTATCTCGAGTACCGGCAATACTGGTTGTTGAGG | |
| MgPR | GTTGGCCGGCCTCGAGGATGAGAATGTTAGATGCCGCGAGTTG | MgPF and MgPR to amplify midgut promoter that was inserted to construct MG QF2 driver plasmid, restriction site XhoI |
| YFPF | GGGCCCGGGATCCACCGGTCGCCACCATGGTGAGCAAGGGCGAGGA | |
| YFPR | GCGGCCGCTACTTGTACAGCTCGTCCA | |
| SgPF | ATCAATGTATCTCGAGGGACTTCGCGTCGGTAGTAG | |
| SgPR | GTTGGCCGGCCTCGAGCGTTTATTCACCTGTGAGCTATGG | YFPF and YFPR to amplify YFP that was inserted to ApaI and NotI sites, then SgPF and SgPR to amplify salivary gland promoter that was inserted to construct SG QF2 driver plasmid at site XhoI |
| MP2-ScopineF | GCGGCCGCGGCTCGAGATGGTGCGATTAAACAGTGCA | |
| MP2-ScopineR | AGATCGACGTCTCGAGTTAGTAGGAGAGTGGAGTAC | MP2-ScopineF and MP2-ScopineR to amplify effectors genes that were inserted to construct QUAS-E plasmid, restriction site XhoI |
| AAPPF | GTACGAAGAGTGCAGCAAGG | |
| AAPPR | TCGATGAGTCCCTCGTCAAG | For RT-PCR: AsAAPP gene |
| PorF | AATGACTCCCAGAAGCAGTG | |
| PorR | ACTTCACTCTTCACACTGCG | For RT-PCR: AsAper1 gene |
| SC1 | GCGGGTTGGATCAATGAG | |
| SC2 | AGTTAGTAGGAGAGTGGA | For RT-PCR: scorpine gene |
| MPF | GTCGAAGCGGCCTGCTAC | |
| MPR | AGATCGACGTTTAGGAGC | For RT-PCR: MP2 gene |
| S7F | CTAACGACACGAAGACCACAAGA | |
| S7R | CAACCTGCAACGAAGCAAAA | For RT-PCR: S7 gene |
| YS1 | AGGACCCTGAAGTTCATCTG | |
| YS2 | CTTCGGGCATGGCGGACTTG | For verification of SG QF2 driver plasmid insertion |
| DM1 | GTGAACTTCCCCTCCGACG | |
| DM2 | TCAGCTTCAGGGCCTTGTG | For verification of MG QF2 driver plasmid insertion |
| E1 | AAAATCCAAAAGAAAATCGATGAGC | |
| E2 | GAGTGGAGTACCACACTTGCAT | For verification of QUAS-MP2-QUAS scorpine effector plasmid insertion |
| SPLNK#1 | CGAAGAGTAACCGTTGCTAGGAGAGACG | |
| SPLNK#2 | GTGGCTGAATGAGACTGGTGTCGAC | |
| pBacRE#1 | CGATATACAGACCGATAAAACACATGCGTC | |
| pBacLE #2 | GCGACTGAGATGTCCTAAATGCAC | Primers for Splinkerette PCR |

**Appendix 1—table 8.** Plasmid injection and screening for transformants.

| Donor plasmid | Helper | # embryos injected | # G0 (number of survivors) | Pools | Pools with positive progeny |
|---|---|---|---|---|---|
| MG QF2 driver plasmid | phsp-pBac | 440 | 35 | 17 | P1 and P2 |
| SG QF2 driver plasmid | phsp-pBac | 500 | 19 | 8 | P1 and P5 |
| QUAS-MP2-Scorpine effector plasmid | phsp-pBac | 547 | 43 | 22 | P1 and P2 |

