## [Editor Report]

This article provides convincing evidence that combining transgenic with paratransgenic approaches can provide a useful tool to reduce malaria transmission by *Anopheles* mosquitoes. The study provides a series of well-designed experiments that will be of interest to malaria and vector control specialists as well as to a broader audience interested in genetic manipulation of insects and paratransgenesis.

---

## [Decision Letter]

**Decision letter after peer review:**

Thank you for submitting your article "Combining transgenesis with paratransgenesis to fight malaria" for consideration by *eLife*. Your article has been reviewed by 2 peer reviewers, one of whom is a member of our Board of Reviewing Editors, and the evaluation has been overseen by Dominique Soldati-Favre as the Senior Editor. The reviewers have opted to remain anonymous.

Essential revisions:

(1) Fitness studies should be performed in Serratia-infected mosquitoes, and in the transgenic-paratransgenic combination.

(2) The hypothesis that inhibition of oocyst development in the Sg/E line is due to mosquitoes ingesting saliva with the blood meal needs to be tested by determining the presence of Scorpine and MP2 protein in the blood bolus.

(3) More data on the stage at which parasites are killed need to be provided, and levels of Scorpine and MP2 need to be determined in paratransgenic mosquitoes.

(4) Limitations of the study need to be more amply discussed.

*Reviewer #2 (Recommendations for the authors):*

The bulk of my review is confined to the 'public review' section.

Below are mostly style elements or requests for points of clarification:

According to Figure 3, you do see an anti-parasite effect just from the addition of wild-type Serratia to the mosquito, correct? The possibility of this, and the fact that it has been found in previous literature, is mentioned but that you also found it in 'this' work does not seem to be explicitly stated.

Line 300-302 would that case still be 'paratransgenesis' then, as implied if there is no transgene?

Line 179 'would have been easily detected' this is a bit difficult to conclude, without any controls.

Line 215 re: mice infections. Authors first talk about % infected (in controls) but then go on to talk about %protected in test experiments. This can be confusing.

Line 220 what is the justification for using median here, rather than mean plus/minus SE? (I get that median of 0% infected sounds better).

116 'this resulted in homozygous lines' needs some more precision – under some scenarios it 'could' result in homozygosity. Similarly, and this is possibly pedantic, but (line 390) 'absence of' does not indicate heterozygosity. It would be better to state all gave 100% inheritance of….consistent with them all being homozygous'.

288 'containment' can have two quite different meanings: containment of parasites or containment of released GM organisms.

Panel B on Figure 1 is redundant and excessive – I initially thought this said something about effector expression in different tissues but it is just whether eyes were red, green, or blue, indicating transgenes. Since this is already done and confirmed molecularly, there is no value in this panel.

On a related note, the splinkerette PCR to identify transgene integrations is well done but the Figure S1 is incorrectly labelled as PCR validation of the plasmid insertion site. Since the primer pair are internal to the transgenes they are not validating the integration sites.

'Significance' section needs a grammar check 'thousand persons yearly' etc. Overall though, I found the manuscript really well written and easy to follow in terms of experimental flow and logic.

[Editors' note: further revisions were suggested prior to acceptance, as described below.]

Thank you for resubmitting your work entitled "Combining transgenesis with paratransgenesis to fight malaria" for further consideration by *eLife*. Your revised article has been evaluated by Dominique Soldati-Favre (Senior Editor) and a Reviewing Editor.

The manuscript has been improved but there are some remaining issues that need to be addressed, as outlined below:

Specifically, the expression levels of Scorpine and MP2 in paratransgenic mosquitoes still need to be tested in order to determine whether levels are correlated to the effects on parasite development.

Moreover, rather than referring to published literature, the presence of Scorpine and MP2 protein in the blood bolus still need to be determined to conclusively demonstrate these effectors affect parasite numbers.

Finally, the discussion of the hurdles to field deployment needs to be more nuanced. For instance, Line 289 concerning "we envision cotton baits soaked with mosquito attractant etc. " is simplistic in its appraisal of the next steps.

---

## [Author Response]

Essential revisions:(1) Fitness studies should be performed in Serratia-infected mosquitoes, and in the transgenic-paratransgenic combination.

These experiments were performed as recommended. As shown in the revised manuscript Appendix 1-Figure A-D, no fitness cost was detected when transgenesis and paratransgenesis are combined.

(2) The hypothesis that inhibition of oocyst development in the Sg/E line is due to mosquitoes ingesting saliva with the blood meal needs to be tested by determining the presence of Scorpine and MP2 protein in the blood bolus.

Ingestion of mosquito saliva during blood feeding and its effect on parasite development has been documented (Luo et al., 2000, Medical Entomology and Zoology 51, 13-20; Hirai et al., 2001, Biochemical and Biophysical Research Communications 287, 859-864). Our previous study showed that human PA-I expressed in salivary gland with the Q-system was ingested together with the saliva and inhibited oocyst development. These studies together with the Western blot showing lack of scorpine and MP2 expression in the midgut of the salivary gland transgenic mosquitoes (Figure 1E) suggest that the oocyst reduction observed in these mosquitoes is due to the effector molecules ingested with the saliva.

(3) More data on the stage at which parasites are killed need to be provided, and levels of Scorpine and MP2 need to be determined in paratransgenic mosquitoes.

Experiments to investigate at which stage parasite are killed were performed as recommended. As shown in Appendix 1-Figure 5 of the revised manuscript, ookinete formation is significantly inhibited by the expression of effector molecules in transgenic mosquitoes and a stronger inhibition is observed when transgenesis is combined with paratransgenesis. These results suggest that the effectors inhibit early stages of *Plasmodium* development in the mosquito midgut.

The effect of scorpine and MP2 on *Plasmodium* development in the midgut of paratransgenic mosquitoes was previously studied in detail, as reported in Conde et al., FEBS Lett 471:165-168 (2000) and in Wang et al., Proc Natl Acad Sci USA 109:12734-12739 (2012) for scorpine and by Vega-Rodríguez et al., Proc Natl Acad Sci USA 111:E492-500 (2014) for MP2.

(4) Limitations of the study need to be more amply discussed.

We thank the reviewers for calling our attention to this point. In response, we have added the following two statements to the discussion.

As effector molecules in the mosquito saliva are injected into the host dermis during blood feeding, the introduction in the field of transgenic mosquitoes that produce non-human proteins in their saliva needs to be considered with much caution.For introduction of bacteria in the field, we envision placing around villages, cotton baits soaked with a mosquito attractant dissolved in sugar and containing suspended bacteria^33^. This is similar to how we introduce bacteria into mosquitoes in the laboratory. The continuous introduction of bacteria into the local mosquito population, combined with the seeding the breeding sites when female mosquitoes lay eggs covered with bacteria^14^, is expected to compensate for the decrease of transmission through mosquito generations.

Reviewer #2 (Recommendations for the authors):The bulk of my review is confined to the 'public review' section.Below are mostly style elements or requests for points of clarification:According to Figure 3, you do see an anti-parasite effect just from the addition of wild-type Serratia to the mosquito, correct? The possibility of this, and the fact that it has been found in previous literature, is mentioned but that you also found it in 'this' work does not seem to be explicitly stated.

This is correct. In our previous study, we also found a similar phenomenon (Wang et al., 2017). In that research, we found around 40% inhibition of *P. falciparum* infection with *Serratia* carrying empty plasmid (HasA), which is similar to what we found here. We added the sentence “As found previously^14^, wild type bacteria also inhibited to some extent, oocyst formation in wild type mosquitoes” (line 195-197).

Line 300-302 would that case still be 'paratransgenesis' then, as implied if there is no transgene?

Good point. We modified the sentence as “The recent finding that a naturally occurring and non-modified *Serratia* can spread through mosquito populations while strongly suppressing *Plasmodium* development^32^, significantly increases the feasibility of moving a paratransgenesis**-**like approach into the field,…” (lines 320-323).

Line 179 'would have been easily detected' this is a bit difficult to conclude, without any controls.

We changed this sentence to “We note that this assay is very sensitive, as one life bacterium in the blood is expected to result in abundant growth during the overnight incubation” (lines 183-184).

Line 215 re: mice infections. Authors first talk about % infected (in controls) but then go on to talk about %protected in test experiments. This can be confusing.

Good point. We changed “protected” to “not infected” throughout.

Line 220 what is the justification for using median here, rather than mean plus/minus SE? (I get that median of 0% infected sounds better).

Due to non-normal distribution, statistics for sporozoite number counts should use non-parametric method (Mann-Whitney's U test), which cannot compare means. Nonetheless, the mean data are provided in the figures.

116 'this resulted in homozygous lines' needs some more precision – under some scenarios it 'could' result in homozygosity. Similarly, and this is possibly pedantic, but (line 390) 'absence of' does not indicate heterozygosity. It would be better to state all gave 100% inheritance of….consistent with them all being homozygous'.

Thank you. The sentence was changed as suggested (Lines 118-121).

288 'containment' can have two quite different meanings: containment of parasites or containment of released GM organisms.

Indeed, the word “containment” was not properly used. This word was omitted in the revised version (line 309).

Panel B on Figure 1 is redundant and excessive – I initially thought this said something about effector expression in different tissues but it is just whether eyes were red, green, or blue, indicating transgenes. Since this is already done and confirmed molecularly, there is no value in this panel.

Thank you for the comment. Figure 1B illustrates how the transgenic lines were first selected, how lines carrying multiple transgenes were selected, and how transgenic lines were built by selecting mosquitoes with the correct combination of eye color. Only after getting enough mosquitoes of the desired lines, we used PCR to confirm the insertions molecularly (this step kills the mosquitoes). Both eye fluorescence and PCR are important in establishing the transgenic lines. Thus, we feel that it is important to keep Figure 1, as it was the primary method used to build the transgenic lines.

On a related note, the splinkerette PCR to identify transgene integrations is well done but the Figure S1 is incorrectly labelled as PCR validation of the plasmid insertion site. Since the primer pair are internal to the transgenes they are not validating the integration sites.

The reviewer is correct. Appendix 1—figure 1 title now reads “plasmid insertion”, not “plasmid insertion site”.

'Significance' section needs a grammar check 'thousand persons yearly' etc. Overall though, I found the manuscript really well written and easy to follow in terms of experimental flow and logic.

Thank you for the suggestion. This was corrected (line 20).

[Editors' note: further revisions were suggested prior to acceptance, as described below.]

The manuscript has been improved but there are some remaining issues that need to be addressed, as outlined below:Specifically, the expression levels of Scorpine and MP2 in paratransgenic mosquitoes still need to be tested in order to determine whether levels are correlated to the effects on parasite development.

As suggested, expression of the multi-effector protein by the bacteria, as well as expression of scorpine and the MP2 peptide by the transgenic mosquitoes was measured by Western blotting of the gut contents one day after a blood meal. This is reported in Appendix 1-Figure 6 of the revised manuscript.

Moreover, rather than referring to published literature, the presence of Scorpine and MP2 protein in the blood bolus still need to be determined to conclusively demonstrate these effectors affect parasite numbers.

As suggested, the presence of scorpine and of the MP2 peptide in the midgut bolus was verified by Western blotting of the midgut contents. This is reported in the Appendix 1-Figure 7 of the revised manuscript.

Finally, the discussion of the hurdles to field deployment needs to be more nuanced. For instance, Line 289 concerning "we envision cotton baits soaked with mosquito attractant etc. " is simplistic in its appraisal of the next steps.

As suggested, the sentences “In the field, we envision the use of attractive sugar feeding stations for Serratia introduction into mosquito populations^33^. Female mosquitoes that acquire the bacteria will seed the breeding sites when they lay eggs^14^” were deleted from the Discussion.